# ORIENT: Submodular Mutual Information Measures for Data Subset Selection under Distribution Shift

**Athresh Karanam**[*]
University of Texas at Dallas
athresh.karanam@utdallas.edu

**Krishnateja Killamsetty**[*]
University of Texas at Dallas
krishnateja.killamsetty@utdallas.edu

**Harsha Kokel**[*]
University of Texas at Dallas
hkokel@utdallas.edu

**Rishabh K Iyer**
University of Texas at Dallas
rishabh.iyer@utdallas.edu

## Abstract

Real-world machine-learning applications require robust models that generalize well to distribution shift settings, which is typical in real-world situations. Domain adaptation techniques aim to address this issue of distribution shift by minimizing the disparities between domains to ensure that the model trained on the source domain performs well on the target domain. Nevertheless, the existing domain adaptation methods are computationally very expensive. In this work, we aim to improve the efficiency of existing supervised domain adaptation (SDA) methods by using a subset of source data that is similar to target data for faster model training. Specifically, we propose ORIENT, a subset selection framework that uses the submodular mutual information (SMI) functions to select a source data subset similar to the target data for faster training. Additionally, we demonstrate how existing robust subset selection strategies, such as GLISTER, GRADMATCH, and CRAIG, when used with a held-out query set, fit within our proposed framework and demonstrate the connections with them. Finally, we empirically demonstrate that SDA approaches like $d$-SNE, CCSA, and standard Cross-entropy training, when employed together with ORIENT, achieve a) faster training and b) better performance on the target data.

## 1   Introduction

The recent success of deep learning frameworks in applications such as image classification [9], speech recognition [20], and object detection [13] stems primarily from the availability of large amounts of labeled data. However, due to enormous labeling costs and the need for specialists in specific domains such as medical imaging, it is not always possible to obtain vast amounts of labeled data in all situations. On the contrary, training deep models on limited amounts of data can lead to poor performance due to overfitting [1]. Consequently, where obtaining large amounts of labeled data is difficult for the target domain, closely related domain (source domain) is used to train the model. However, this may result in a deep model with suboptimal performance on target domain as a result of the distribution shift [47, 5, 4, 50], i.e., change of data distribution from source domains to target domains. A change in the data distribution often renders the features learned by the model on the source domain irrelevant for the target domain.

To address the problem of distribution shift, many domain adaptation [54, 41] and domain generalization [40] techniques have been proposed in recent years. Domain adaptation methods assume that some target domain information is available (usually target data), whereas domain generalization methods do not. Domain adaptation (DA) methods can be categorized into unsupervised [15, 12, 32, 52]

---

[*]equal contribution

36th Conference on Neural Information Processing Systems (NeurIPS 2022).

(using unlabeled target data), semi-supervised [16, 59, 43, 48] (using labeled and unlabeled target data), and supervised [39, 38, 58, 37, 19] (using labeled target data) methods. UDA methods that do not require any labeled target data assume access to large volumes of unlabeled target data. When providing the same amount of target data, SDA methods are more effective than UDA methods [39]. As it is not difficult to obtain a few samples (as less as 2 examples per class) of labeled target data in practice, SDA methods are an attractive approach in scenarios with limited target data. Hence, in this work we are explicitly focused on the SDA setting.

Recently, various effective SDA methods are proposed [39, 58, 37]. However, they are usually compute intensive. For example, using one of the state-of-the-art SDA method, $d$-SNE [58], to train a ResNet50 model [18] on the Office-Home dataset [53] with 3022 samples of the a source data and 338 samples a target data for 300 epochs takes $> 18$ hours using a `GTX 1080 Ti GPU`. To put that in perspective, training a ResNet50 model on the larger CIFAR100 dataset with 45000 training samples for 300 epochs on the same machine for standard cross-entropy loss takes only 14 hours. The increase in training time not only increases energy consumption and CO2 emissions but also restricts the usefulness of SDA methods in resource-constrained environments. We address this problem by seeking an answer to the following question: ***Can we improve the efficiency of SDA methods by training on subsets of source data to ensure faster adaptation and reduction in training time?***

To this end, we propose ORIENT, a subset selection framework that utilizes Submodular Mutual Information (SMI) measures [21, 17] to select subsets of source data that are similar to target data for training. Our key insight is that by training the model on data points that are similar to target data, we can speed up the training. The ORIENT framework, illustrated in Figure 1, complements existing SDA methods and can be effectively combined with any of them, enabling us to achieve a reduction in training times, energy costs, and CO2 emissions.

The key contributions of our paper include: (1) ORIENT, an SMI-based subset selection framework for efficient and effective supervised domain adaptation. (2) In Section 3, we illustrate that ORIENT unifies three existing subset selection methods: CRAIG [35], GLISTER [25], and GRAD-MATCH [24]. Specifically, these methods fit into ORIENT's framework when used with a held-out validation set from the target domain. (3) We empirically show the effectiveness of ORIENT when used with existing SDA methods like CCSA [39], $d$-SNE [58], and Standard Cross-entropy training on two real-world datasets: Office-31 [42] and Office-Home [53]. Our experiments also demonstrate that model learn better class discrimination features with ORIENT, resulting in better classification performance.

## 1.1 Related work

**SDA** When the labeled training data is scarce for a domain $\tau$ (or a task), a powerful way to boost performance is by pretraining a model on a related domain and fine-tuning it in the target domain [14, 60, 8]. I SDA techniques have been investigated for various distribution shift settings where the target domain ($\tau^t = \{X^t, Y^t\}$) has different marginal or conditional distributions than the source domain ($\tau^s = \{X^s, Y^t\}$). In this work, we consider *covariate shift* [47], wherein the marginal distibution is different $\left(\text{i.e. } P(X^s) \neq P(X^t)\right)$, but the conditional distribution remains the same $\left(\text{i.e. } P(Y^s|X^s) = P(Y^t|X^t) = P(Y|X)\right)$. A notable body of work for covariate shift SDA has focused on identifying a latent feature space that is domain-invariant [39, 38, 58, 37, 19]. This line of work train a network consisting of a feature extractor and a classifier end-to-end. The feature extractor learns a non-linear transformation of the samples from different domains to a shared latent space, and the classifier learns to assign class labels. Different researchers have proposed different optimization objectives and loss functions to encourage domain confusion and class separability.

Tzeng et al. [51] designed a domain-confusion loss to optimize for domain invariance and match the distribution over classes in the source domain to the soft label in the target domain. Motiian et al. [39] propose a classification and contrastive semantic alignment loss (CCSA). CCSA uses contrastive loss with Siamese Network to encourage the same class samples from different distributions to be nearby in the latent space (semantic alignment loss), and the different class samples from different distributions to be far apart (separation loss). Few-shot adversarial domain adaptation (FADA) [38] proposes to learn similar latent space with adversarial training. Domain-adaptation using stochastic neighborhood embedding ($d$-SNE) [58] proposes to maximizes the smallest distance between the samples of different classes and minimizes the largest distance between the samples of the same class while being domain invariant. Another approach, Second- or Higher-order Transfer of Knowledge

(So-HoT) [29], aims to align the with-in class scatters and maintain separation of between-class scatters. Finally, Morsing et al. [37] propose domain adaptation using graph embedding (DAGE) and show that CCSA and $d$-SNE can also be expressed as graph embedding methods [19].

**Subset Selection methods** Submodular function [33, 11] is one of the effective approach of data subset selection which is used in variety of applications [6]. For example, it has been employed for efficient training in speech recognition [56, 55], machine translation [28], computer vision [22], supervised classification [35, 25, 24], semi-supervised classification [26], active learning methods [57, 46, 2, 25], and hyper-parameter optimization [27]. Another widely used method for selecting subsets is coreset construction. The coresets [10] are weighted subsets of the data that approximate the semantic characteristics of the entire dataset (e.g., loss, marginal probability, etc.). A number of recent coreset selection-based methods [36, 25, 24, 26] have demonstrated the promise to efficiently and robustly train deep models. Coresets are used for other deep learning applications like continual learning, active learning and data summarization [7, 49].

Subset selection plays a significant role in robust learning under realistic scenarios, such as imbalances or rare classes, out-of-distribution(OOD) data, or redundancy. Axelrod et al. [3] propose a simple cross-entropy based data selection methods for effective domain adaptation in statistical machine learning. Kothawade et al. [30] employed SMI measures for active learning in realistic scenarios tackling imbalance, OOD, and redundancy. PRISM [31] employed SMI measures for targeted data summarization. GLISTER [25] and GRADMATCH [24] shows that using a clean held-out validation set with SMI mitigates the label noise and class imbalance in the training set. Mirzasoleiman et al. [36] showed the data samples with clean labels cluster together, whereas the ones with noisy labels spread out in the gradient space, thus making k-medoid clustering an effective method for reducing noise in labeled data. Furthermore, Killamsetty et al. [26] tackles OOD and imbalance in the unlabeled data in semi-supervised learning setting through data subset selection. We show how these previous subset selection strategies including a modified version of CRAIG [35] were also using instantiations of SMI measures in Section 3.

## 2 Preliminaries

In this section, we introduce submodular functions, SMIs and SDA loss functions.

**Submodular Functions:** Let $D$ be a set of $n$ data points $D = \{1, \cdots, n\}$ and $f : 2^D \to \mathbb{R}$ be a set function returning real-value for any subset of set $D$. A function $f$ is *submodular* [11] if $f(A \cup \{x\}) - f(A) \geq f(B \cup \{x\}) - f(B), \forall x \in D, \forall A \subseteq B \subseteq D$ and $x \notin B$. Monotone Submodular functions, when maximized for a cardinality constraint using a simple greedy algorithm, admit a constant factor of $1 - \frac{1}{e}$. This can be done in near-linear time using a stochastic-greedy algorithm.

**Submodular Mutual Information (SMI):** The SMI between two sets $A, B \subseteq D$, *instantiated* with a submodular function $f$, is defined as $I_f(A; B) = f(A) + f(B) - f(A \cup B)$ [21]. In this work, we aim to select data points $A \subset D$ that maximize the SMI between $A$ and $B$. Intuitively, this allows selection of data points that are *similar* to $B$ while being diverse.

**SDA Loss Functions:** For supervised domain adaptation, specialized loss functions are introduced as described in the previous section. Here we highlight two state-of-the-art domain adaptation loss functions, CCSA and $d$-SNE, which are used in our experiments.

In CCSA, the classifier is modeled as a composition of two functions–$h \circ g$. Here, $g : X \to Z$ is a feature extractor that transforms the input from the feature space $X$ to an embedding space $Z$, and $h : Z \to Y$ is a predictor function. The CCSA loss for supervised domain adaptation is defined as,

$$\mathcal{L}_{CCSA}(h \circ g) = \mathcal{L}_{\text{CE}}(h \circ g) + \mathcal{L}_{SA}(g) + \mathcal{L}_S(g),$$

where $\mathcal{L}_{\text{CE}}(h \circ g)$ is the *cross-entropy loss* for multi-class classification, $\mathcal{L}_{SA}$ is a *semantic alignment loss* encouraging the samples from different domains but same label to map nearby in the embedding space, and $\mathcal{L}_S$ is a *separation loss* encouraging the samples from different domains and different labels to map far away in the embedding space.

The $d$-SNE loss function for SDA is defined as

$$\mathcal{L}_{d\text{-SNE}}(h \circ g) = \tilde{\mathcal{L}}(g) + \alpha \mathcal{L}_{\text{CE}}^s(h \circ g) + \beta \mathcal{L}_{\text{CE}}^t(h \circ g),$$

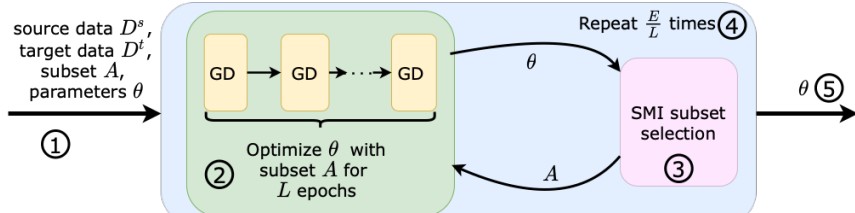

Figure 1: Illustration of ORIENT framework. **(1)** Given the target data $D^t$ and source data $D^s$, a subset $A \subseteq D^s$ and model parameters $\theta$ are randomly initialized. **(2)** Model parameters $\theta$ are optimized for $L$ epochs on the subset $A$ with any gradient descent (GD) based method. **(3)** Subset $A$ is updated using the SMI measure and current model parameters $\theta$ after every $L^{th}$ epoch. **(4)** Model is trained for $E$ epochs, with subset selection after every $L$ interval. **(5)** The final model parameters are returned after $E$ epochs.

| **Name** | $f(A)$ | $I_f(A; D^t)$ |
|---|---|---|
| FLMI | $\sum\limits_{i \in D^t} \max\limits_{j \in A} S_{ij}$ | $\sum\limits_{i \in D^t} \max\limits_{j \in A} S_{ij} + \eta \sum\limits_{i \in A} \max\limits_{j \in D^t} S_{ij}$ |
| LOGDETMI | $\log \det(S_A)$ | $\log \det(S_A) - \log \det(S_A - \eta^2 S_{A,D^t} S_{D^t}^{-1} S_{A,D^t}^T)$ |
| GCMI | $\sum\limits_{i \in A, j \in D^s} S_{ij} - \lambda \sum\limits_{i,j \in A} S_{ij}$ | $2\lambda \sum\limits_{i \in A, j \in D^t} S_{ij}$ |
| COM | Appendix Eq. 3 | $\eta \sum\limits_{i \in A} \psi\left(\sum\limits_{j \in D^t} S_{ij}\right) + \sum\limits_{j \in D^t} \psi\left(\sum\limits_{i \in A} S_{ij}\right)$ |

Table 1: Instantiations of submodular mutual information functions [31].

where $\mathcal{L}_{\text{CE}}^s$ and $\mathcal{L}_{\text{CE}}^t$ are the cross-entropy loss on the source and the target domain examples, respectively, and $\tilde{\mathcal{L}}$ minimizes the largest distance between the samples from different domains with same label and maximizes the smallest distance between the samples from different domain with different label in the embedding space. SDA loss functions are further elaborated in Section A.4.

## 3 ORIENT

For our SDA setting, we assume a small amount of labeled data (as less as 2 examples per class) is available from the target domain $\tau^t$ and sufficient labeled data is available from the source domain $\tau^s$. We denote the source dataset as $D^s = (x_i, y_i)_{i=1}^{|D^S|}$ and the target dataset as $D^t = (x_i, y_i)_{i=1}^{|D^t|}$. The source dataset is used for training the model $\mathcal{M}$ using an SDA loss function $\mathcal{L}$. We will introduce other necessary notations in the remainder of the paper, as necessary. A complete list of notations is presented in Appendix (A.1).

First, we extend the SMI measure to incorporate data subsets from different domains. For a source domain subset $A \subseteq D^s$ and the target domain data set $D^t$, we denote SMI measure as $I_f(A; D^t)$. Essentially, assuming that both the domains form a set $\mathcal{D}$, and $A \subseteq D^s \subset \mathcal{D}$ and $D^t \subset \mathcal{D}$. With $S$ denoting the similarity matrix defined over a subset $A$ and $D^t$, Table 1 presents the mathematical expression of the SMI functions for different instantiations of submodular functions ($f$) [31]. We defer the readers to Kothawade et al. [31] for more details. Note that the FLMI, GCMI and COM only need the pairwise similarities of the data points in $A$ and $D^t$. Consequently, the similarity kernel is of size $|A| \times |D^t|$, making it very efficient to optimize.

For efficient SDA, we want to select a subset $A$ of the source domain $\tau^s$ such that training a model on $A$ results in a classifier that is proficient on the target domain $\tau^t$. To this effect, we use the gradients $\chi$ of the current model to represent the data points and use it to compute the similarity matrix. Specifically, we define the pairwise similarity between two data points as the cosine similarity

between the gradients,

$$S_{ij} = \frac{\chi_i\,\chi_i}{|\chi_i|\,|\chi_j|} \tag{1}$$

$$\text{where, } \chi_i = \nabla_\theta \mathcal{L}(x_i, y_i) \text{ and } \chi_j = \nabla_\theta \mathcal{L}(x_j, y_j)$$

Given a set size $b$, we select a *diverse* subset $A$ of the source data $D^s$ that is *similar* to the target data $D^t$ by maximizing the following SMI measure,

$$\underset{A \subseteq D^s, |A| \leq b}{\arg\max}\ I_f(A; D^t) \tag{2}$$

$I_f(A; D^t)$ is an instance of cardinality constrained monotone submodular maximization for all SMI functions except for LogDetMI. Therefore, we can achieve a $1 - \frac{1}{e}$ approximation using the lazy greedy algorithm [34]. Even though LogDetMI is not submodular, previous works [31, 30] have reported good empirical performance using the lazy greedy algorithm for maximization of the LogDetMI function. Following the footsteps of Kothawade et al. [31, 30], we also use the lazy greedy algorithm to maximize the LogDetMI function. However, as the number of parameters in the deep learning model can be extremely high, our gradients $\chi$ can be very high dimensional. Following the success of targeted subset selection approaches, GLISTER [25], SIMILAR [30], CRAIG [36] and BADGE [2], we circumvent this problem by using last-layer gradient approximations to represent the data point in the similarity matrix $S$. Further, we select a new subset every $L$ epochs as the model is trained, and the subsets chosen are adapted accordingly.

**Algorithm:** We present the complete training procedure of ORIENT in Algorithm 1. We first randomly initialize the training subset $A$ and the model parameters $\theta$ in **line 1** and **2**, respectively. For each epoch, we update the model parameters by optimizing a gradient-descent (GD) based loss $\mathcal{L}$ on the current training subset $A$ in **line 4**. After a fixed interval of $L$ epochs, we update the subset $A$ (**line 5–11**). To do this, we compute the gradients $\chi^s$ and $\chi^t$ for all the datapoints in $D^s$ and $D^t$, in **line 6** and **7**, respectively. Next, we use the gradients to compute the similarity matrix $S$ in **line 8**, as described in Equation 1. In **line 9** we instantiate the SMI function $I_f$ with $S$ and update the subset $A$ in **line 10** using Equation 2. Our implementation is made available online[2].

---

**Algorithm 1** ORIENT

---

**Input:** source domain data $D^s$, query data $D^t$, batch size $b$, total epochs $E$, subset selection interval $L$, SMI function $I_f$.

**Output:** Final model parameters $\theta$

1: $A \leftarrow \text{RandomSubset}(D^s)$ ▷ *Randomly initialize a subset*
2: $\theta$ ▷ *initial model parameters*
3: **for** epoch $e$ in $E$ **do** ▷ *for each epoch*
4:     $\theta \leftarrow \text{mini-batch-gd}(\theta, \mathcal{L}(A))$ ▷ *mini-batch gradient descent*
5:     **if** $e \mod L == 0$ **then** ▷ *perform subset selection every $L$ epochs*
6:        $\chi^s \leftarrow \nabla_\theta \mathcal{L}(D^s)$ ▷ *compute gradients for source data*
7:        $\chi^t \leftarrow \nabla_\theta \mathcal{L}(D^t)$ ▷ *compute gradients for query data*
8:        $S \leftarrow \text{SIMILARITY}(\chi^s, \chi^t)$ ▷ *compute the similarity matrix*
9:        Instantiate $I_f$ with $S$
10:       $A \leftarrow \underset{A \subseteq D^s, |A| \leq b}{\arg\max}\ I_f(A; D^t)$ ▷ *update subset $A$*

---

Figure 1 illustrates the ORIENT workflow. The general framework of ORIENT can incorporate any gradient decent (GD) based SDA technique. It can train any model with loss function $\mathcal{L}$ by using a subset of the source data, selected every $L$ epochs. In our experiments, we demonstrate the effectiveness of our method in conjunction with standard cross-entropy loss as well as two state-of-the-art domain adaptation losses, CCSA [39] and d-SNE [58].

### Connections to Previous Work

ORIENT generalizes three approaches, GLISTER [25], GRADMATCH [24], and a slightly adapted version of CRAIG [35], which were initially proposed for efficient and robust subset selection.

---

[2]Anonymized link: https://github.com/athresh/orient

Specifically, these subset selection approaches maximize a submodular function corresponding to a different instantiations of SMI functions for subset selection. Following theorems summarize the connection of SMI functions with these subset selection approaches.

**Theorem 1** *When the outer level loss of the discrete bi-level optimization problem of* GLISTER *is hinge loss, logistic loss, and perceptron loss, then the optimization problem becomes an instance of Concave over Modular (COM) SMI function.*

**Theorem 2** *When the optimization problem of* GRADMATCH *is used to match gradients of a held-out validation set, it becomes an instance of the summation of GCMI and a diversity function.*

**Theorem 3** *When the optimization problem of* CRAIG *is adapted to match gradients of a held-out validation set, it becomes an instance of the FLMI function with $\eta = 0$.*

We present the proofs of these theorems in Appendix (A.3). In conclusion, SMI measures have been used in previous subset selection strategies for robust learning to deal with the class imbalance and noisy labels in training data sets and achieved comparable performance to current state-of-the-art robust supervised learning approaches.

## 4   Experimental Evaluation

Our experiments explicitly aim at answering the following questions,

Q1:  Does the proposed ORIENT approach substantially reduce the training time while maintaining comparable performance to training on complete source dataset?

Q2:  Can the proposed ORIENT approach augment the existing domain adaptation approaches?

We evaluate our proposed approach on two domain adaptation datasets: Office-31 [42] and Office-Home [53]. Office-31 dataset consists of $4110$ images of 31 object categories from three different domains: Amazon (A), DSLR (D), and Webcam (W). For our experiments we used all the images from the source domain in the training set ($D^s$) and two examples of each category in the query set ($D^t$). Office-Home dataset consists of $10812$ images of 65 object categories from four different domains: Art (A), Clipart (C), Product (P), and Real-world (R). Here, the target data ($D^t$) consists of about $20\%$ of the images from the target domain. We use $L = 20$, that is, we sample the subset after every 20 epochs and use $b = 0.3\%$, that is, we sample $30\%$ of the source domain for training.

To answer the first question, we use ORIENT to train a ResNet50 architecture using the stochastic gradient descent (SGD) algorithm with the momentum of 0.9 and the weight decay ratio of $5e4$. We compare our approach against three baselines: **Full**—training on the source data and the target data, i.e. $D^s \cup D^t$; **Random**—a random subset of the source dataset and the target set, i.e. $R \cup D^t, R = \mathrm{RandomSubset}(D^s)$; and **CRAIG**—a coreset of the source dataset, i.e $\mathrm{coreset}(D^t)$, without any information of the target dataset. We use standard categorical cross-entropy loss function (for multi-class classification) and five different instantiations of the submodular mutual information function: Facility Location Mutual Information (**ORIENT (FLMI)**), Graph Cut Mutual Information (**ORIENT (GCMI)**), Log Determinant Mutual Information (**ORIENT (LDMI)**), GLISTER (**ORIENT (G)**), and GradMatch (**ORIENT (GM)**).

| | | A $\rightarrow$ D | A $\rightarrow$ W | D $\rightarrow$ A | D $\rightarrow$ W | W $\rightarrow$ A | W $\rightarrow$ D |
|---|---|---|---|---|---|---|---|
| CCSA | Full | 0.78 | 0.72 | 0.55 | **0.93** | 0.55 | **0.97** |
| | Random | 0.76 | 0.72 | 0.54 | 0.81 | 0.54 | 0.92 |
| | ORIENT | 0.77 | **0.76** | 0.55 | 0.89 | 0.55 | **0.96** |
| d-SNE | Full | 0.77 | 0.69 | 0.53 | **0.93** | 0.54 | **0.98** |
| | Random | 0.76 | 0.68 | 0.53 | 0.86 | 0.53 | 0.94 |
| | ORIENT | 0.78 | 0.71 | **0.55** | 0.90 | 0.56 | **0.97** |

Table 2: Test accuracy for office-31 with SDA methods

Figure 2 presents the scatter plot of the speed up against the prediction accuracy of Office-31 dataset using cross-entropy loss.. In all our experiments, we use $0.3$ fraction of the source data as the subset

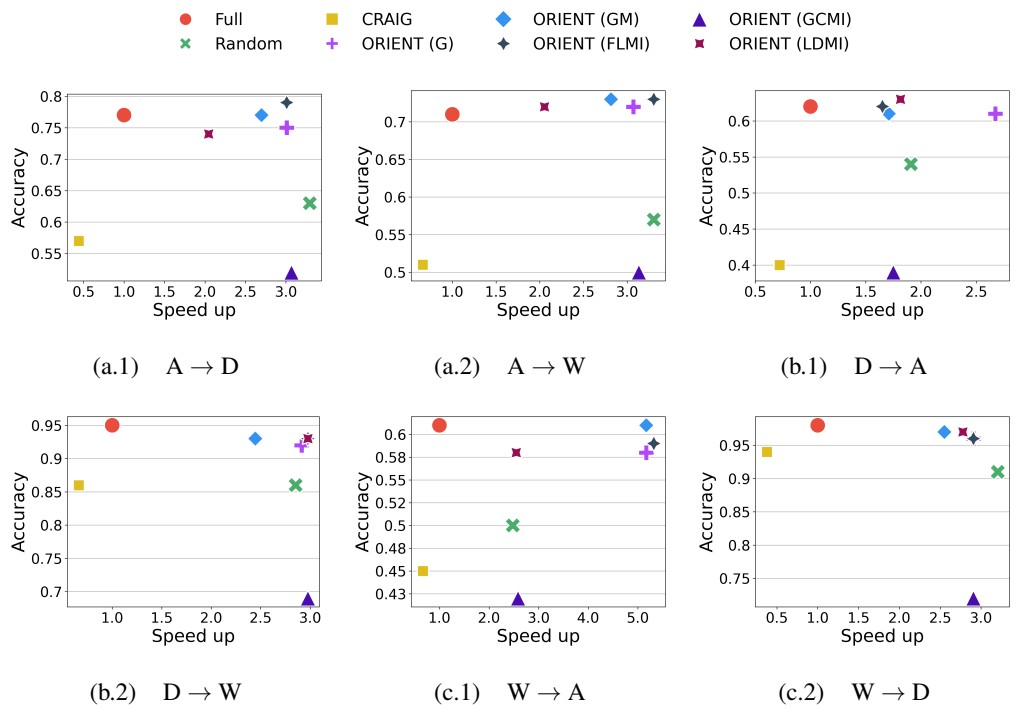

| | | | | | | | |
|---|---|---|---|---|---|---|---|
| Full | Random | CRAIG | ORIENT (G) | ORIENT (GM) | ORIENT (FLMI) | ORIENT (GCMI) | ORIENT (LDMI) |

(a.1)  A → D

(a.2)  A → W

(b.1)  D → A

(b.2)  D → W

(c.1)  W → A

(c.2)  W → D

Figure 2: Speed up vs prediction accuracy on three domains of Office31 dataset: Amazon (A), DSLR (D), and Webcam (W). $X$-axis represents the speed up by the model, i.e. the ratio of the time taken to train on complete source dataset (Full) to the time taken by the model. $Y$-axis represents the prediction accuracy of the model on the target domain.

| | | R → P | R → C | P → R | P → C | C → R | C → P | A → P | A → R | A → C | R → A | P → A | C → A |
|---|---|---|---|---|---|---|---|---|---|---|---|---|---|
| **CCSA** | Full | 0.74 | **0.55** | 0.62 | 0.46 | 0.56 | 0.67 | 0.70 | **0.64** | 0.48 | 0.57 | **0.49** | 0.41 |
| | Random | 0.71 | 0.47 | 0.62 | 0.45 | 0.54 | 0.66 | 0.69 | 0.57 | 0.48 | 0.5 | 0.47 | 0.45 |
| | ORIENT | **0.78** | **0.54** | **0.65** | **0.5** | **0.61** | **0.71** | 0.71 | **0.65** | **0.51** | **0.59** | **0.54** | **0.47** |
| **d-SNE** | Full | 0.77 | **0.53** | **0.62** | **0.50** | **0.60** | 0.71 | **0.72** | 0.63 | 0.49 | **0.52** | 0.44 | 0.40 |
| | Random | 0.75 | 0.50 | 0.60 | 0.45 | 0.57 | 0.69 | 0.68 | 0.59 | 0.49 | 0.46 | 0.43 | 0.40 |
| | ORIENT | 0.77 | 0.52 | 0.63 | 0.50 | 0.60 | 0.71 | **0.71** | 0.61 | 0.51 | **0.52** | 0.44 | 0.42 |

Table 3: Test accuracy for Office-Home with SDA methods

and see $> 2.5\times$ speed up. Figure 3 presents the convergence curves of the Office-Home dataset using cross-entropy loss. It is evident from these charts that our proposed approach ORIENT can achieve better or comparable performance to the Full training in significantly less amount of time, indicating better efficiency. This analysis helps us answer our Q1. ORIENT reduces the training time substantially, the speed up achieved is reciprocal to the fraction of the subset used, without trading the performance. Rather, in some combinations, we observe that ORIENT outperforms Full.

Figure 4 presents the bar plots of the speed up when ORIENT(FLMI) is used in conjunction with $d$-SNE loss function on Office-31 and Office-Home datasets. Here, we see a consistent $3\times$ speed up as compared to full model training with $d$-SNE loss. These bar plots demonstrate that our proposed approach ORIENT can augment existing domain adaptation approaches and substantially reduce the training times. Additionally, Tables 2 and 3 present the test accuracy while using $d$-SNE and CCSA loss in conjunction with ORIENT(FLMI) on Office-31 and Office-Home datasets, respectively. It's evident that augmenting existing domain adaptation approaches with ORIENT achieves better or comparable performance to the Full training. These two observations help us answer our Q2 in the affirmative. Precise values of prediction accuracy and training time for all the experiments are provided in Appendix (A.5). Further, in Appendix Appendix (A.7) and Appendix (A.8), we analyze the effect of different subset sizes and subset sampling frequencies.

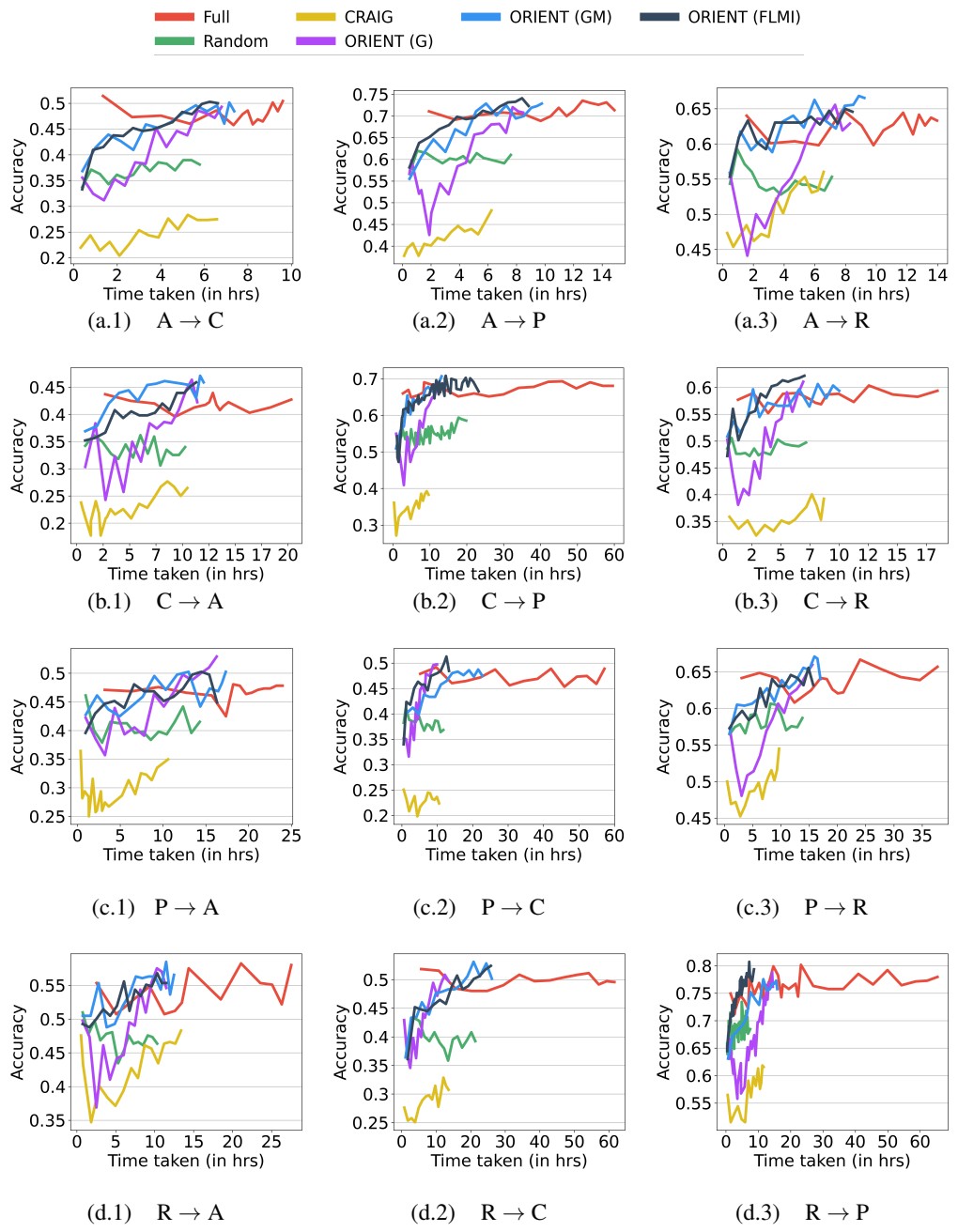

Figure 3: Convergence curves on four domains of Office-Home dataset: Art (A), Clipart (C), Product (P), and Real World (R). $X$-axis presents the training time in hours and $Y$-axis presents the prediction accuracy on the target domain.

Figure 5 presents the GradCam [45] class-activation maps of trained models on the Office-Home dataset ($P \rightarrow R$ setting) using $d$-SNE loss for both Full and ORIENT (FLMI). These activation maps show that the model trained with the ORIENT framework learned better class discrimination features than the model trained with Full. This might explain why the ORIENT framework performs better sometimes than Full.

**Comparison of different SMI functions:** Domain adaptation results on Office31 (Figure 2) and OfficeHome (Figure 3) datasets show that ORIENT(GCMI) performs suboptimally compared to ORIENT using other SMI functions in terms of target domain accuracy. Although ORIENT(LDMI)

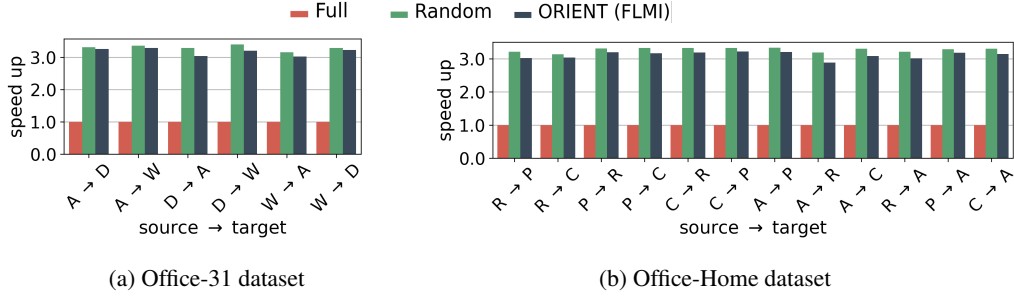

(a) Office-31 dataset         (b) Office-Home dataset

Figure 4: Speed up achieved by combining d-SNE with ORIENT

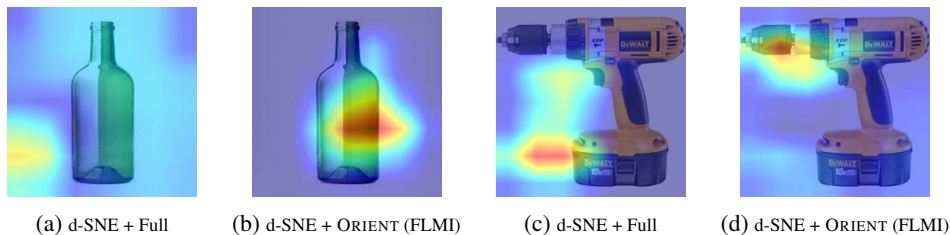

(a) d-SNE + Full    (b) d-SNE + ORIENT (FLMI)    (c) d-SNE + Full    (d) d-SNE + ORIENT (FLMI)

Figure 5: GradCam [45] activation maps of the models learned using d-SNE + Full and d-SNE + ORIENT (FLMI) on the Office-Home dataset with "Product" as the source domain and "Real World" as the target domain. As evidenced by class activation maps, the ORIENT framework enabled the model to learn more effective class discriminative features than Full data training.

achieves reasonable target domain accuracy, it is computationally expensive and does not achieve the best performance-speedup trade-off. In comparison to the ORIENT using remaining SMI functions, i.e., ORIENT(GM), ORIENT(G), and ORIENT(FLMI), ORIENT(FLMI) consistently achieves the best performance versus speed-up trade-off. We further present synthetic experiments to provide intuitions on how different SMI functions selects data subsets in Appendix A.6

To summarize, our experiments on two adaptation datasets suggest that our proposed approach ORI-ENT can substantially reduce the training time while maintaining comparable performance to training on complete source data. Additionally, our experiments using $d$-SNE and CCSA loss functions suggest that ORIENT can be used in conjunction with existing SDA methods to achieve significant speed ups in training time, while maintaining or improving the prediction accuracy. In particular, two instantiations of our proposed approach, ORIENT(FLMI) and ORIENT(GM) consistently achieve comparable or better performance compared to Full training while being $\sim 3\times$ faster to train.

## 5 Conclusion, Limitations, and Broader Impact

We introduce ORIENT, a subset selection framework based on SMI functions for supervised domain adaptation. The submodularity of SMI functions allows us to use scalable greedy algorithms to select the data subsets efficiently. In addition, we demonstrate how ORIENT is a unified framework that integrates previous approaches based on subset selection for robust learning. Empirically, we show that ORIENT is very effective for SDA. Specifically, it achieves $\sim 3\times$ speed up over existing SDA approaches like $d$-SNE and CCSA while achieving comparable or better performance. Our findings confirm that ORIENT has a significant social impact by making existing SDA algorithms significantly faster and more energy-efficient, reducing CO2 emissions and energy consumption incurred during training, thus contributing to a Green AI [44]. The main limitation of our work is that although ORIENT significantly reduces the training time, it requires more memory to store the similarity kernel required for subset selection. This makes running ORIENT harder on devices with low memory.

## Acknowledgments and Disclosure of Funding

AK acknowledges the support by the NIH grant R01HD101246, HK gratefully acknowledges the support of the ARO award W911NF2010224. RI and KK would like to acknowledge support from NSF Grant Number IIS-2106937, a gift from Google Research, and the Adobe Data Science Research award. Authors would like to acknowledge Dr. Sriraam Natarajan for helpful discussions and support. The views and conclusions contained herein are those of the authors and should not be interpreted as necessarily representing the official policies or endorsements, either expressed or implied, of the ARO, NIH, NSF, Google Research, Adobe Data Science or the U.S. government.

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
