# A Appendix

## A.1 Notations

Table 4 summarize the notations used in this paper.

| | Notation | Description |
|---|---|---|
| DATA | $\tau, \tau^s, \tau^t$ | domain or task |
| | superscript $^s$ | source domain |
| | superscript $^t$ | target domain |
| | $X$ | Feature space |
| | $Y$ | Label space |
| | $X$ | Embedding space |
| | $P(X), P(X^s), P(X^t)$ | Marginal distributions |
| | $P(Y\|X)$ | Conditional distribution |
| | $D$ | Set of data points |
| | $A, B$ | Subset of $D$ |
| | $D^s, D^t$ | Datasets |
| SMI | $f$ | Sub modular function |
| | $I_f$ | SMI function |
| | $S$ | Similarity matrix |
| ORIENT | $\mathcal{L}$ | Loss function |
| | $b$ | Batch size |
| | $E$ | Total training epochs |
| | $L$ | Epoch interval |
| SDA | $g$ | feature extractor |
| | $h$ | predictor function |
| | $\circ$ | Composition |

Table 4: Notations

## A.2 Details on Concave over Modular Mutual Information

$f_\eta(A)$ is a restricted submodular function [31] defined over sets $V, V'$ and $\psi$ is a concave function. Let $n$ be the size of set $A$.

$$
\begin{aligned}
f_\eta(A) =& \eta \sum_{i \in V'} \max\left(\psi\left(\sum_{j \in A \cap V} S_{ij}\right), \psi\left(\sqrt{n} \sum_{j \in A \cap V'} S_{ij}\right)\right) \\
&+ \sum_{i \in V} \max\left(\psi\left(\sum_{j \in A \cap V'} S_{ij}\right), \psi\left(\sqrt{n} \sum_{j \in A \cap V} s_{ij}\right)\right)
\end{aligned}
\tag{3}
$$

## A.3 Proof of the Technical Results

**Theorem 4** *When the outer level loss of the discrete bi-level optimization problem of* GLISTER *is hinge loss, logistic loss, and perceptron loss, then the optimization problem becomes an instance of maximization of Concave over Modular (COM) SMI function.*

PROOF Given, training loss $L_T$, validation loss $L_V$, training set $\mathcal{D}$, subset $\mathcal{S}$, subset size $k$, and validation set $\mathcal{V}$, the objective of GLISTER can be written as follows:

$$
\begin{aligned}
&\min_{\mathcal{S} \subseteq \mathcal{D}, |\mathcal{D}|=k} L_V(\theta^*, \mathcal{V}) \\
&\text{where } \theta^* = \arg\min_\theta L_T(\theta, \mathcal{S})
\end{aligned}
\tag{4}
$$

Loss on the subset denoted by $L_T(\theta, \mathcal{S})$ is the summation of the losses of individual data samples in the subset. i.e., $L_T(\theta, \mathcal{S}) = \sum_{(x,y)\in\mathcal{S}} L_T(\theta, x, y)$. Using the one-step gradient approximation of GLISTER, the above optimization problem can be written as:

$$\min_{\mathcal{S}\subseteq\mathcal{D}, |\mathcal{D}|=k} L_V(\theta - \eta\nabla_\theta L_T(\theta, \mathcal{S}), \mathcal{V})$$

$$\min_{\mathcal{S}\subseteq\mathcal{D}, |\mathcal{D}|=k} L_V(\theta - \eta \sum_{(x,y)\in\mathcal{S}} \nabla_\theta L_T(\theta, x, y), \mathcal{V}) \tag{5}$$

where $\eta$ is the learning rate.

We can convert the above minimization problem to a maximization problem as following:

$$\max_{\mathcal{S}\subseteq\mathcal{D}, |\mathcal{D}|=k} -L_V(\theta - \eta \sum_{(x,y)\in\mathcal{S}} \nabla_\theta L_T(\theta, x, y), \mathcal{V}) \tag{6}$$

Using the Proof of Theorem-1 in GLISTER [25], the optimization problem in Equation 6 for different validation losses can be written as follows:

*Case 1* When $L_V$ is hinge loss or perceptron loss, the optimization problem of GLISTER is:

$$\max_{\mathcal{S}\subseteq\mathcal{D}, |\mathcal{D}|=k} \sum_{i=1}^{|\mathcal{V}|} \min(0, C_i + \sum_{j\in\mathcal{S}} \hat{g}_{ij})$$

$$\text{where } \sum_{j\in\mathcal{S}} \hat{g}_{ij} \geq 0 \tag{7}$$

We can write the above optimization problem as,

$$\max_{\mathcal{S}\subseteq\mathcal{D}, |\mathcal{D}|=k} |\mathcal{V}|C_i + \sum_{i=1}^{|\mathcal{V}|} \min(-C_i, \sum_{j\in\mathcal{S}} \hat{g}_{ij}) \tag{8}$$

$$\max_{\mathcal{S}\subseteq\mathcal{D}, |\mathcal{D}|=k} \sum_{i=1}^{|\mathcal{V}|} \min(-C_i, \sum_{j\in\mathcal{S}} \hat{g}_{ij}) \tag{9}$$

Note that in the above equation, $\min(C, x)$ is a concave function in x and $\sum_{j\in\mathcal{S}} \hat{g}_{ij}$ is a non-negative modular function. Hence, the above formulation is submodular and is an instance of concave over modular function.

*Case 2* When $L_V$ is logistic loss, the optimization problem of GLISTER is:

$$\max_{\mathcal{S}\subseteq\mathcal{D}, |\mathcal{D}|=k} \sum_{i=1}^{|\mathcal{V}|} C - \log\left(1 + C_i \exp\left(\alpha \sum_{j\in\mathcal{S}} \hat{g}_{ij}\right)\right) \tag{10}$$

Note that in the above equation, $-\log(1 + C \exp -x)$ is concave in x and $\sum_{j\in\mathcal{S}} \hat{g}_{ij}$ is modular. Hence, the above formulation of set function is submodular and is an instance of concave over modular function. In our setting, we use labeled target data $D^t$ as the validation set. Furthermore, the above given formulations of GLISTER corresponds to instantiation of maximization of COM MI function $(\eta \sum_{i\in A} \psi(\sum_{j\in D^t} S_{ij}) + \sum_{j\in D^t} \psi(\sum_{i\in A} S_{ij}))$ with $\eta = 0$.

**Theorem 5** *When the optimization problem of* GRADMATCH *with equal sample weights is used to match gradients of a held-out validation set, it becomes an instance of maximization of summation of GCMI and a diversity function.*

PROOF Given, training loss $L_T$, validation loss $L_V$, training set $\mathcal{D}$, and validation set $\mathcal{V}$. Let us denote loss of $i^{th}$ sample in the training dataset as $L_T^i(\theta)$ and the loss of $j^{th}$ sample in the validation dataset as $L_V^j(\theta)$.

The objective of GRADMATCH with equal sample weights can be written as follows:

$$\min_{\mathcal{S}\subseteq\mathcal{D},|\mathcal{S}|=k}\left\|\frac{1}{k}\sum_{i\in\mathcal{S}}\nabla_\theta L_T^i(\theta_t)-\frac{1}{|\mathcal{V}|}\sum_{j\in\mathcal{V}}\nabla_\theta L_V^j(\theta)\right\|^2 \tag{11}$$

Without the loss of generality we assumed sample weights to be 1 in the above equation.

We can convert this to a maximization problem as follows:

$$\max_{\mathcal{S}\subseteq\mathcal{D},|\mathcal{S}|=k}-\left\|\frac{1}{k}\sum_{i\in\mathcal{S}}\nabla_\theta L_T^i(\theta_t)-\frac{1}{|\mathcal{V}|}\sum_{j\in\mathcal{V}}\nabla_\theta L_V^j(\theta)\right\|^2 \tag{12}$$

$$\max_{\mathcal{S}\subseteq\mathcal{D},|\mathcal{S}|=k}\frac{-1}{k^2}\left\|\sum_{i\in\mathcal{S}}\nabla_\theta L_T^i(\theta)\right\|^2+\frac{-1}{|\mathcal{V}|^2}\left\|\sum_{j\in\mathcal{V}}\nabla_\theta L_V^j(\theta_t)\right\|^2+2\frac{1}{k|\mathcal{V}|}\sum_{i\in\mathcal{S},j\in\mathcal{V}}\nabla_\theta L_T^i(\theta)^T\cdot\nabla_\theta L_V^j(\theta)$$
$$\tag{13}$$

In the above equation second term is not dependent on $\mathcal{S}$ and can be ignored during optimization. Following which the above optimization problem can be written as:

$$\max_{\mathcal{S}\subseteq\mathcal{D},|\mathcal{S}|=k}\frac{-1}{k^2}\left\|\sum_{i\in\mathcal{S}}\nabla_\theta L_T^i(\theta)\right\|^2+2\frac{1}{k|\mathcal{V}|}\sum_{i\in\mathcal{S},j\in\mathcal{V}}\nabla_\theta L_T^i(\theta)^T\cdot\nabla_\theta L_V^j(\theta) \tag{14}$$

Note that in our setting, labeled target data $D^t$ is used as the validation set. In the above equation, the second term corresponds to GCMI function($2\lambda\sum_{i\in A,j\in D^t}S_{ij}$) with $\lambda=1$.

Expanding on the first term we have,

$$\frac{-1}{k^2}\left\|\sum_{i\in\mathcal{S}}\nabla_\theta L_T^i(\theta)\right\|^2=\frac{-1}{k^2}\sum_{i\in\mathcal{S}}\left\|\nabla_\theta L_T^i(\theta)\right\|^2-\frac{2}{k^2}\sum_{i,j\in\mathcal{S}|i\neq j}\nabla_\theta L_T^i(\theta)^T\cdot\nabla_\theta L_T^j(\theta) \tag{15}$$

The function given in Equation 15 is a diversity function.

Hence, the optimization problem of GRADMATCH with equal sample weights when matched with validation set is a instance of maximization of summation of the GCMI function and diversity function.

**Theorem 6** *When the optimization problem of* CRAIG *is adapted to match gradients of a held-out validation set, it becomes an instance of maximization of the FLMI function with $\eta=0$.*

PROOF Given, training loss $L_T$, validation loss $L_V$, training set $\mathcal{D}$, and validation set $\mathcal{V}$. Let us denote loss of $i^{th}$ sample in the training dataset as $L_T^i(\theta)$ and the loss of $j^{th}$ sample in the validation dataset as $L_V^j(\theta)$.

The objective of CRAIG matching gradients of a held-out validation set is as follows:

$$\max_{\mathcal{S}\subseteq\mathcal{D},|\mathcal{S}|=k}\sum_{i\in\mathcal{V}}\max_{j\in\mathcal{S}}\nabla_\theta L_V^i(\theta)^T\cdot\nabla_\theta L_T^j(\theta) \tag{16}$$

Note that in our setting, labeled target data $D^t$ is used as the validation set. The set function in the above equation corresponds to FLMI function($\sum_{i\in D^t}\max_{j\in A}S_{ij}+\eta\sum_{i\in A}\max_{j\in D^t}S_{ij}$) with $\eta=0$.

Hence, the optimization problem of CRAIG adapted to match gradients of a held-out validation set is an instance of maximization of the FLMI function with $\eta=0$.

## A.4 SDA Loss

**CCSA:** In CCSA, the classifier is modeled as a composition of two functions–$h \circ g$. Here, $g : X \to Z$ is a feature extractor that transforms the input from the feature space $X$ to an embedding space $Z$, and $h : Z \to Y$ is a predictor function. Let $X_y^s$ and $X_y^t$ denote the source and the target domain samples with label $y \in Y$, respectively. The semantic alignment loss ($\mathcal{L}_{SA}$) for CCSA is defined as,

$$\mathcal{L}_{SA}(g) = \sum_{y \in Y} d\left(P\left(g\left(X_y^s\right)\right), P\left(g\left(X_y^t\right)\right)\right),$$

where $d(.)$ indicates a distance measure between the distributions of $X_y^s$ and $X_y^t$ in the embedding space and $P(.)$ indicates the distribution. $\mathcal{L}_{SA}$ encourages the samples from different domains and the same label to map nearby in the embedding space. The separation loss ($\mathcal{L}_S$) is defined as

$$\mathcal{L}_S(g) = \sum_{a,b \in Y | a \neq b} k\left(P\left(g\left(X_a^s\right)\right), P\left(g\left(X_b^t\right)\right)\right),$$

where $k$ is a similarity measure that returns a higher value when the distribution of $X_a^s$ and $X_b^t$ is close in the embedding space. Hence, $\mathcal{L}_S$ encourages the samples from different domains and different labels to map far away in the embedding space. Overall the CCSA loss is defined as a combination of cross-entropy loss, semantic alignment loss, and separation loss,

$$\mathcal{L}_{CCSA}(h \circ g) = \mathcal{L}_{\text{CE}}(h \circ g) + \mathcal{L}_{SA}(g) + \mathcal{L}_S(g).$$

The distance measure ($d$) in the semantic alignment loss and the similarity measure ($k$) in the separation loss are computed as average of pairwise similarities and distances between all the samples from the source and the target domain, respectively. Further assumptions on $d$ and $k$ lends them into a well known contrastive loss function (cf. Motiian et al. [39] for details).

**$d$-SNE:** Instead of minimizing the average of distances between all pairs of samples, Xu et al. [58], in $d$-SNE, proposes to minimize the largest distance of the samples from different source with same label, and maximizing the smallest distance of the samples from different source and different labels. Let $D_y^s$ denote the subset of source data with label $y$, then the loss function $\tilde{L}(g)$ is defined as,

$$\tilde{L}(g) = \sum_{x_j \in D^t} \left( \sup_{x \in D_{y_j}^s} \left\{ a \mid a \in d\left(g(x), g(x_j)\right) \right\} - \inf_{x \in D^s \backslash D_{y_j}^s} \left\{ b \mid b \in d\left(g(x), g(x_j)\right) \right\} \right).$$

The complete $d$-SNE loss is defined as a combination of $\tilde{L}$ and cross-entropy loss on source and target domain.

$$\mathcal{L}_{d\text{-SNE}}(h \circ g) = \tilde{\mathcal{L}}(g) + \alpha \mathcal{L}_{\text{CE}}^s(h \circ g) + \beta \mathcal{L}_{\text{CE}}^t(h \circ g).$$

## A.5 Additional Results

Figure 6 presents the GradCam [45] class-activation maps of trained models on the Office-Home dataset ($P \to R$ setting) using $d$-SNE loss for both Full and ORIENT (FLMI). These activation maps show that the model trained with ORIENT framework learn effective class discriminative features more consistently than Full.

We present the test accuracy of models trained using cross-entropy loss on a combination of source and target domain data, $D^s \cup D^t$, of Office-31 and Office-Home datasets in 5 and 9, respectively. We performed 3 runs of each experiment using different initial training data subset each time. We see that ORIENT(FLMI) and ORIENT(GM) perform similar to Full and outperform Random across all experiments. Tables 6 and 10 show the training times for these settings. We see that all instantiations of ORIENT except ORIENT(L) achieve $\sim 3\times$ speed-up compared to Full.

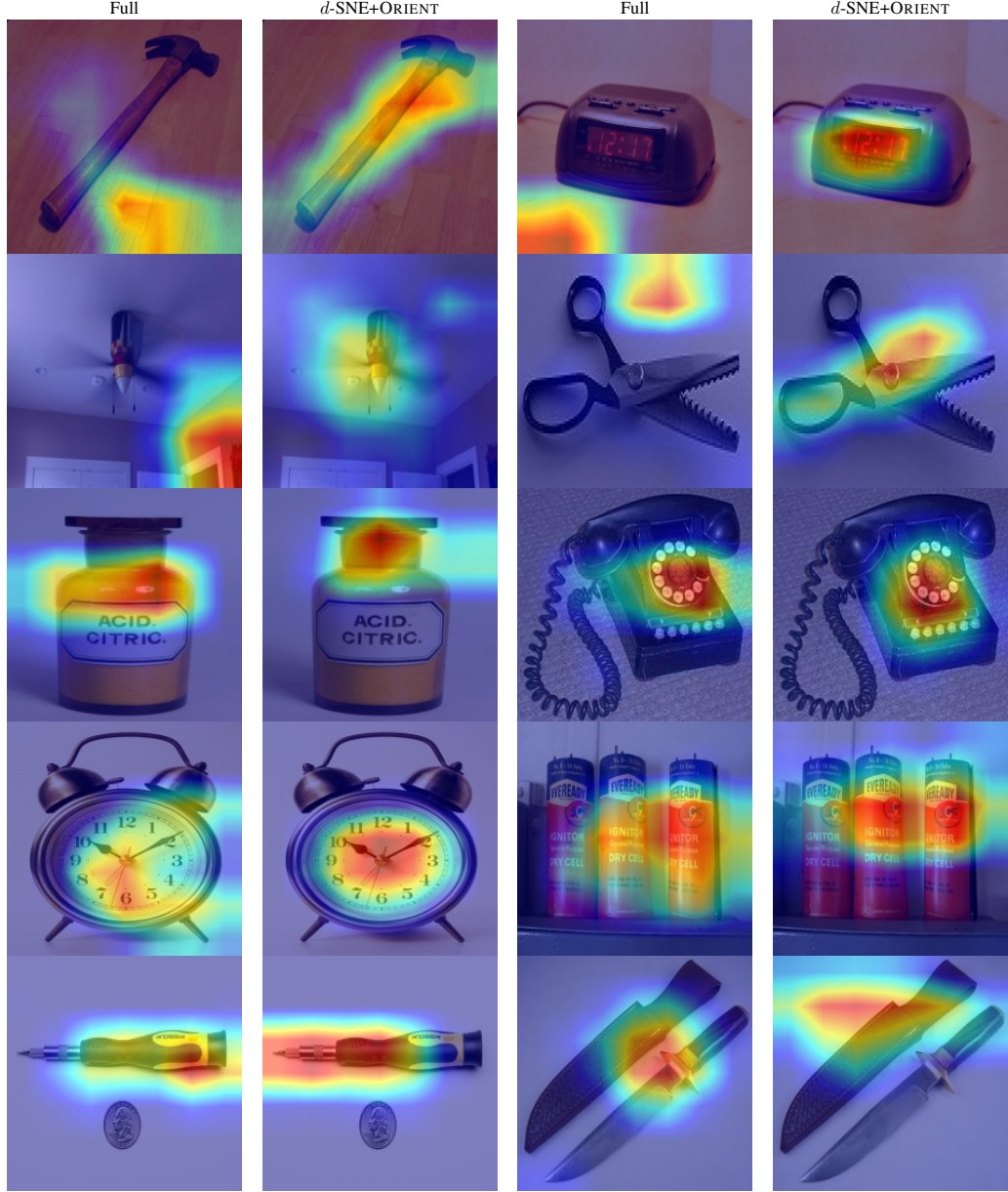

Figure 6: GradCam [45] activation maps of the models learned using $d$-SNE + Full and $d$-SNE + ORIENT (FLMI) on the Office-Home dataset with "Product" as the source domain and "Real World" as the target domain. As evidenced by class activation maps, the ORIENT framework enabled the model to learn more effective class discriminative features than Full data training consistently.

|  | A → D | A → W | D → A | D → W | W → A | W → D |
|---|---|---|---|---|---|---|
| Full | $0.77 \pm 0.02$ | $0.71 \pm 0.02$ | $0.62 \pm 0.01$ | $0.95 \pm 0.01$ | $0.61 \pm 0.01$ | $0.98 \pm 0.01$ |
| Random | $0.63 \pm 0.03$ | $0.57 \pm 0.04$ | $0.54 \pm 0.02$ | $0.86 \pm 0.02$ | $0.5 \pm 0.03$ | $0.91 \pm 0.02$ |
| ORIENT (FLMI) | $0.79 \pm 0.02$ | $0.73 \pm 0.01$ | $0.62 \pm 0.01$ | $0.93 \pm 0.01$ | $0.59 \pm 0$ | $0.96 \pm 0.01$ |
| ORIENT (GC) | $0.52 \pm 0.12$ | $0.5 \pm 0.08$ | $0.39 \pm 0.07$ | $0.69 \pm 0.11$ | $0.42 \pm 0.09$ | $0.72 \pm 0.08$ |
| ORIENT (L) | $0.74 \pm 0.03$ | $0.72 \pm 0.02$ | $0.63 \pm 0.01$ | $0.93 \pm 0.01$ | $0.58 \pm 0.01$ | $0.97 \pm 0.01$ |
| ORIENT (G) | $0.75 \pm 0.01$ | $0.72 \pm 0.01$ | $0.61 \pm 0.01$ | $0.92 \pm 0.01$ | $0.58 \pm 0.01$ | $0.96 \pm 0$ |
| ORIENT (GM) | $0.77 \pm 0.02$ | $0.73 \pm 0.02$ | $0.61 \pm 0.01$ | $0.93 \pm 0.01$ | $0.61 \pm 0.01$ | $0.97 \pm 0.01$ |

Table 5: Test accuracy on Office-31 dataset.

| | A → D | A → W | D → A | D → W | W → A | W → D |
|---|---|---|---|---|---|---|
| Full | $3.13 \pm 0.02$ | $3.04 \pm 0.02$ | $1.47 \pm 0.52$ | $1.37 \pm 0$ | $1.81 \pm 1.12$ | $1.25 \pm 0.01$ |
| Random | $0.95 \pm 0.01$ | $0.92 \pm 0$ | $0.77 \pm 0.04$ | $0.48 \pm 0.13$ | $0.73 \pm 0.01$ | $0.39 \pm 0$ |
| ORIENT (FLMI) | $1.04 \pm 0.09$ | $0.82 \pm 0.18$ | $0.89 \pm 0.01$ | $0.46 \pm 0.01$ | $0.34 \pm 0.01$ | $0.43 \pm 0$ |
| ORIENT (GC) | $1.02 \pm 0.02$ | $0.97 \pm 0$ | $0.84 \pm 0.01$ | $0.46 \pm 0$ | $0.7 \pm 0.02$ | $0.43 \pm 0.01$ |
| ORIENT (L) | $1.53 \pm 0$ | $1.48 \pm 0.02$ | $0.81 \pm 0.01$ | $0.46 \pm 0$ | $0.71 \pm 0.01$ | $0.45 \pm 0$ |
| ORIENT (G) | $1.04 \pm 0.02$ | $0.99 \pm 0$ | $0.55 \pm 0.11$ | $0.47 \pm 0$ | $0.35 \pm 0.01$ | $0.43 \pm 0$ |
| ORIENT (GM) | $1.16 \pm 0.03$ | $1.08 \pm 0.01$ | $0.86 \pm 0.22$ | $0.56 \pm 0.02$ | $0.35 \pm 0$ | $0.49 \pm 0.01$ |

Table 6: Training time (in hours) for 300 epochs on Office-31 dataset

| | | A → D | A → W | D → A | D → W | W → A | W → D |
|---|---|---|---|---|---|---|---|
| CCSA | Full | $0.78 \pm 0$ | $0.72 \pm 0$ | $0.55 \pm 0$ | $0.93 \pm 0$ | $0.55 \pm 0$ | $0.97 \pm 0$ |
| | Random | $0.76 \pm 0.01$ | $0.72 \pm 0.01$ | $0.54 \pm 0.02$ | $0.81 \pm 0.02$ | $0.54 \pm 0.01$ | $0.92 \pm 0.02$ |
| | ORIENT (G) | $0.78 \pm 0.03$ | $0.73 \pm 0.02$ | $0.55 \pm 0.04$ | $0.92 \pm 0.01$ | $0.56 \pm 0$ | $0.97 \pm 0.01$ |
| | ORIENT (GM) | $0.78 \pm 0.01$ | $0.74\pm$ | $0.51 \pm 0.04$ | $0.88 \pm 0.02$ | $0.55 \pm 0.02$ | $0.97 \pm 0.02$ |
| | ORIENT (FLMI) | $0.77 \pm 0$ | $0.76 \pm 0.01$ | $0.55 \pm 0$ | $0.89 \pm 0.02$ | $0.55 \pm 0.02$ | $0.96 \pm 0.01$ |
| d-SNE | Full | $0.77 \pm 0$ | $0.69 \pm 0$ | $0.53 \pm 0.01$ | $\mathbf{0.93 \pm 0}$ | $0.54 \pm 0$ | $0.98 \pm 0$ |
| | Random | $0.76 \pm 0.02$ | $0.68 \pm 0.02$ | $0.53 \pm 0.02$ | $0.86 \pm 0.02$ | $0.53 \pm 0.01$ | $0.94 \pm 0.02$ |
| | ORIENT (G) | $0.73 \pm 0$ | $0.69 \pm 0.01$ | $0.52 \pm 0.02$ | $\mathbf{0.93 \pm 0.01}$ | $0.54 \pm 0$ | $0.97 \pm 0.01$ |
| | ORIENT (GM) | $0.76 \pm 0.01$ | $0.69\pm$ | $0.52 \pm 0.04$ | $0.89 \pm 0.02$ | $0.54 \pm 0.02$ | $0.97 \pm 0.01$ |
| | ORIENT (FLMI) | $0.78 \pm 0$ | $0.71 \pm 0.01$ | $\mathbf{0.55 \pm 0}$ | $0.90 \pm 0.02$ | $0.56 \pm 0.02$ | $0.97 \pm 0.01$ |

Table 7: Test accuracy on Office-31 dataset with SDA methods

| | | A → D | A → W | D → A | D → W | W → A | W → D |
|---|---|---|---|---|---|---|---|
| CCSA | Full | $24.11 \pm 0.02$ | $27.81 \pm 1.0$ | $7.16 \pm 0.1$ | $7.84 \pm 0.01$ | $10.46 \pm 0.01$ | $10.54 \pm 1.15$ |
| | Random | $14.71 \pm 2.5$ | $14.64 \pm 0.19$ | $3.37 \pm 0.02$ | $2.64 \pm 0.39$ | $4.11 \pm 0.08$ | $3.25 \pm 0.77$ |
| | ORIENT (G) | $16.59 \pm 0.42$ | $15.53 \pm 1.2$ | $3.63 \pm 0$ | $3.45 \pm 0.48$ | $4.53 \pm 0.21$ | $4.14 \pm 0.77$ |
| | ORIENT (GM) | $17.11 \pm 1.11$ | $15.6\pm$ | $3.14 \pm 0.95$ | $4.1 \pm 0.34$ | $5.33 \pm 0.6$ | $5.21 \pm 1.29$ |
| | ORIENT (FLMI) | $13.93 \pm 8.85$ | $13.24 \pm 8.47$ | $3.79 \pm 0.65$ | $3.53 \pm 1.3$ | $5.34 \pm 1.27$ | $5.21 \pm 1.25$ |
| d-SNE | Full | $11.57 \pm 0.02$ | $8.33 \pm 0.5$ | $2.01 \pm 0.1$ | $2.21 \pm 0.1$ | $2.18 \pm 0.1$ | $3.42 \pm 0.2$ |
| | Random | $3.49 \pm 0.05$ | $2.48 \pm 0.01$ | $0.61 \pm 0.02$ | $0.65 \pm 0.013$ | $0.69 \pm 0.01$ | $1.04 \pm 0.02$ |
| | ORIENT (G) | $3.53 \pm 0.05$ | $2.5072 \pm 0.02$ | $0.66 \pm 0.05$ | $0.69 \pm 0.01$ | $0.72 \pm 0.01$ | $1.06 \pm 0.01$ |
| | ORIENT (GM) | $3.56 \pm 0.01$ | $2.56 \pm 0.01$ | $0.68 \pm 0.02$ | $0.71 \pm 0.03$ | $0.73 \pm 0.03$ | $1.10 \pm 0.02$ |
| | ORIENT (FLMI) | $3.55 \pm 0.01$ | $2.53 \pm 0.04$ | $0.66 \pm 0.01$ | $0.69 \pm 0.01$ | $0.72 \pm 0.02$ | $1.06 \pm 0.02$ |

Table 8: Training time in hours on Office-31 with SDA methods

| | R→P | R→C | P→R | P→C | C→R | C→P | A→P | A→R | A→C | R→A | P→A | C→A |
|---|---|---|---|---|---|---|---|---|---|---|---|---|
| Full | 0.79 | 0.5 | 0.62 | **0.49** | 0.59 | 0.68 | 0.71 | 0.63 | **0.5** | **0.58** | 0.48 | 0.43 |
| Random | 0.69 | 0.39 | 0.58 | 0.37 | 0.5 | 0.58 | 0.61 | 0.55 | 0.38 | 0.46 | 0.42 | 0.34 |
| ORIENT (G) | 0.77 | 0.5 | **0.66** | 0.48 | 0.61 | 0.67 | 0.71 | 0.63 | 0.49 | 0.55 | **0.53** | 0.42 |
| ORIENT (GM) | 0.76 | 0.5 | 0.64 | 0.47 | 0.59 | 0.71 | **0.73** | **0.67** | 0.48 | 0.57 | 0.5 | **0.46** |
| ORIENT (FLMI) | 0.79 | **0.52** | 0.63 | 0.48 | **0.62** | **0.69** | 0.72 | 0.65 | **0.5** | 0.55 | 0.45 | **0.46** |

Table 9: Test accuracy on Office-Home dataset

| | R→P | R→C | P→R | P→C | C→R | C→P | A→P | A→R | A→C | R→A | P→A | C→A |
|---|---|---|---|---|---|---|---|---|---|---|---|---|
| Full | 44.28 | 61.58 | 37.72 | 57.2 | 18.47 | 59.6 | 14.86 | 13.96 | 9.61 | 27.55 | 23.97 | 20.32 |
| Random | 6.92 | 21.34 | 13.8 | 11.72 | 7.16 | 17.5 | 7.6 | 7.11 | 5.79 | 10.32 | 14.3 | 10.27 |
| ORIENT (G) | 14.41 | 13.65 | 15.57 | 9.95 | 6.93 | 12.61 | 8.46 | 8.29 | 6.8 | 11.49 | 16.28 | 11.42 |
| ORIENT (GM) | 15.88 | 26.15 | 17 | 22.78 | 10.02 | 13.18 | 9.77 | 9.23 | 7.38 | 12.48 | 17.34 | 12.02 |
| ORIENT (FLMI) | 7.78 | 25.84 | 15.73 | 13.33 | 7.02 | 19.97 | 8.85 | 8.45 | 6.62 | 11.47 | 16.33 | 11.31 |

Table 10: Training time(in hours) for 300 epochs on Office-Home dataset

| | | R→P | R→C | P→R | P→C | C→R | C→P | A→P | A→R | A→C | R→A | P→A | C→A |
|---|---|---|---|---|---|---|---|---|---|---|---|---|---|
| CCSA | Full | 0.74 | **0.55** | 0.62 | 0.46 | 0.56 | 0.67 | 0.70 | 0.64 | 0.48 | 0.57 | 0.49 | 0.41 |
| | Random | 0.71 | 0.47 | 0.62 | 0.45 | 0.54 | 0.66 | 0.69 | 0.57 | 0.48 | 0.5 | 0.47 | 0.45 |
| | ORIENT(G) | **0.78** | 0.54 | 0.63 | **0.5** | 0.59 | 0.7 | **0.72** | 0.62 | 0.5 | 0.57 | 0.49 | **0.5** |
| | ORIENT(GM) | 0.75 | 0.5 | 0.63 | 0.48 | 0.59 | 0.69 | 0.69 | 0.63 | 0.49 | 0.51 | 0.5 | 0.46 |
| | ORIENT(FLMI) | **0.78** | 0.54 | **0.65** | **0.5** | **0.61** | **0.71** | 0.71 | **0.65** | **0.51** | **0.59** | **0.54** | 0.47 |
| d-SNE | Full | **0.77** | 0.53 | 0.62 | **0.50** | **0.60** | 0.71 | **0.72** | **0.63** | 0.49 | 0.52 | 0.44 | 0.40 |
| | Random | 0.75 | 0.50 | 0.60 | 0.45 | 0.57 | 0.69 | 0.68 | 0.59 | 0.49 | 0.46 | 0.43 | 0.40 |
| | ORIENT(G) | 0.75 | 0.51 | 0.62 | 0.49 | 0.59 | 0.7 | **0.72** | 0.62 | 0.5 | **0.54** | 0.44 | 0.41 |
| | ORIENT(GM) | 0.76 | **0.52** | 0.62 | **0.50** | 0.59 | 0.70 | 0.71 | 0.61 | 0.49 | 0.51 | **0.46** | **0.42** |
| | ORIENT(FLMI) | **0.77** | **0.52** | **0.63** | **0.50** | **0.60** | 0.71 | 0.71 | 0.61 | **0.51** | 0.52 | 0.44 | **0.42** |

Table 11: Test accuracy on Office-Home with SDA methods

| | | R→P | R→C | P→R | P→C | C→R | C→P | A→P | A→R | A→C | R→A | P→A | C→A |
|---|---|---|---|---|---|---|---|---|---|---|---|---|---|
| CCSA | Full | 17.73 | 17.53 | 18.23 | 14.93 | 17.25 | 16.42 | 10.32 | 9.45 | 16.13 | 17.67 | 8.29 | 11.35 |
| | Random | 5.57 | 5.17 | 4.74 | 4.87 | 5.57 | 4.62 | 3.48 | 3.12 | 4.25 | 5.88 | 2.52 | 3.45 |
| | ORIENT (G) | 6.01 | 6.94 | 5.78 | 5.15 | 6.61 | 5.92 | 3.71 | 3.41 | 4.61 | 6.13 | 2.87 | 4.12 |
| | ORIENT (GM) | 5.73 | 5.69 | 5.15 | 5.39 | 6.12 | 5.34 | 3.78 | 3.85 | 4.66 | 6.27 | 3.17 | 3.95 |
| | ORIENT (FLMI) | 6.12 | 6.13 | 5.73 | 6.17 | 5.56 | 5.99 | 3.94 | 3.34 | 4.89 | 6.72 | 2.78 | 4.23 |
| d-SNE | Full | 16.84 | 16.68 | 17.42 | 9.55 | 17.05 | 9.60 | 6.64 | 9.94 | 6.29 | 18.81 | 11.69 | 11.27 |
| | Random | 5.23 | 5.31 | 5.26 | 2.87 | 5.12 | 2.88 | 1.99 | 3.11 | 1.90 | 5.85 | 3.55 | 3.41 |
| | ORIENT (G) | 5.48 | 5.61 | 5.42 | 2.98 | 5.32 | 2.96 | 2.11 | 3.48 | 1.95 | 6.21 | 3.63 | 3.53 |
| | ORIENT (GM) | 6.08 | 6.01 | 5.50 | 3.09 | 5.51 | 3.12 | 2.17 | 3.65 | 2.08 | 6.62 | 3.81 | 3.93 |
| | ORIENT (FLMI) | 5.56 | 5.48 | 5.44 | 3.01 | 5.34 | 2.98 | 2.07 | 3.44 | 2.04 | 6.23 | 3.67 | 3.58 |

Table 12: Training time(in hours) on Office-Home with SDA methods

We present test accuracy of models trained using SDA loss on source domain data, $D^s$, of Office-31 dataset in 7. Again, we see that all instantiations of ORIENT perform similar to Full while outperforming Random. Table 8 shows the training times for this setting. Again, we see that all instantiations of ORIENT achieve $\sim 2.5\times$ speed-up compared to Full.

Finally, Table 11 presents the test accuracy of models trained using SDA loss on source domain data, $D^s$, of Office-Home dataset. Here, we see that instantiations of ORIENT perform substantially better than Full and Random in some experiments and perform similar to Full in the rest. In particular, ORIENT(FLMI) outperforms Full in the settings of $R \rightarrow P, C \rightarrow R, C \rightarrow P$ and $P \rightarrow A$. Table 12 presents the corresponding training times. Once again, we see that instantiations of ORIENT achieve $\sim 3\times$ speed-up compared to Full.

## A.6 Synthetic Experiments

To provide better intuition into how different SMI functions select data subsets, we present a comparison of the subset selected for two toy datasets in Figures 7 and 8. Figure 7a and 8a presents the synthetic dataset with 5000 samples in the source domain. The target domain consists of the 500 data points highlighted with a different color. The query set is of size 50. We present the subset selected by ORIENT with Facility Location Mutual Information (**ORIENT (FLMI)**), GradMatch (**ORIENT (GM)**), GLISTER (**ORIENT (G)**), Log Determinant Mutual Information (**ORIENT**

**(LDMI)**), and Graph Cut Mutual Information (**ORIENT (GCMI)**). From the results, it is evident that the FLMI function selects sample sources closer to the target domain than other SMI functions. In contrast, the GM function selects representative samples from the source domain. Our results show that the G function selects samples near the decision boundaries of the source domain data. The GCMI function selects samples from a source domain that are very similar and clustered together. In synthetic results, we found that the LDMI function tends to prioritize the selection of data samples from certain classes compared to others.

## A.7 Analysis of Data Subset Size

Table 13 presents the target domain accuracies achieved using different subset sizes on the office-31 dataset using d-SNE loss, and Table 14 presents the training times taken by using different subset sizes on the office-31 dataset using d-SNE loss. As seen, on reducing the subset size from $0.3$ fraction to $0.1$, model experiences loss of accuracy but gains in terms of training time. Hence, there is a trade-off between training time and accuracy of the model. Higher fraction of the data would lead to better performance in accuracy but also require more training time. Where as, lower fraction would require less training time but might result in lower accuracy.

|  | Fraction | A $\to$ D | A $\to$ W | D $\to$ A | D $\to$ W | W $\to$ A | W $\to$ D |
|---|---|---|---|---|---|---|---|
| Full | 1.0 | 0.77 | 0.69 | 0.53 | 0.93 | 0.54 | 0.98 |
| Random | 0.1 | 0.65 | 0.58 | 0.43 | 0.62 | 0.48 | 0.72 |
| Random | 0.3 | 0.76 | 0.68 | 0.53 | 0.86 | 0.53 | 0.94 |
| ORIENT(FLMI) | 0.1 | 0.70 | 0.67 | 0.50 | 0.77 | 0.48 | 0.92 |
| ORIENT(FLMI) | 0.3 | 0.78 | 0.71 | 0.55 | 0.90 | 0.56 | 0.97 |
| ORIENT(G) | 0.1 | 0.75 | 0.60 | 0.42 | 0.70 | 0.48 | 0.89 |
| ORIENT(G) | 0.3 | 0.76 | 0.66 | 0.50 | 0.87 | 0.52 | 0.96 |

Table 13: Comparison of test accuracy for office-31 with d-SNE loss function and different fractions of subset selection.

|  | Fraction | A $\to$ D | A $\to$ W | D $\to$ A | D $\to$ W | W $\to$ A | W $\to$ D |
|---|---|---|---|---|---|---|---|
| Full | 1.0 | 11.57 | 8.33 | 2.01 | 2.21 | 2.18 | 3.42 |
| Random | 0.1 | 1.18 | 0.83 | 0.20 | 0.22 | 0.23 | 0.34 |
| Random | 0.3 | 3.49 | 2.48 | 0.61 | 0.65 | 0.69 | 1.04 |
| ORIENT (FLMI) | 0.1 | 1.25 | 0.89 | 0.25 | 0.26 | 0.26 | 0.38 |
| ORIENT (FLMI) | 0.3 | 3.55 | 2.53 | 0.66 | 0.69 | 0.72 | 1.06 |
| ORIENT (G) | 0.1 | 1.21 | 0.87 | 0.25 | 0.26 | 0.25 | 0.37 |
| ORIENT (G) | 0.3 | 3.53 | 2.51 | 0.66 | 0.69 | 0.72 | 1.06 |

Table 14: Training time in hours on Office-31 with d-SNE loss function and different fractions of subset selection.

## A.8 Analysis of $L$ for subset selection

We present comparison of target domain accuracy achieved by ORIENT (FLMI) for different L values of 5, 10, 20, 40 on Office 31 (A->D) using d-SNE loss and 30% subset in Table 15. Note that smaller the value of $L$ is, greater the frequency of subset selection. Results demonstrate that using $L = 5, 10$ (i.e., more frequent subset selection) results in higher training time with no improvement in accuracy. Whereas using $L = 40$(i.e., less frequent subset selection) results in lower target domain accuracy with not much significant improvement in training time. Hence, we used $L = 20$ in our experiments.

## A.9 Analysis of time taken

Additionally, we present the convergence curve of training time against the validation loss in Fig. 9. As different methods use different losses, the absolute values of loss are not directly comparable. But we still like to present these plots to show that even though the Full starts with reasonable

Figure 7: Subsets selected by different instantiations of ORIENT on a synthetic dataset. (a) Synthetic data - We sample 50 examples from the target distributions for the query set. (b) ORIENT (FLMI) selects samples close to the target distribution. (c) ORIENT (GM) selects representative samples from the source domain. (d) ORIENT (G) selects samples near the decision boundaries of the source domains. (e) ORIENT(LDMI) prioritizes selection of data samples from certain classes over others (f) ORIENT(GCMI) selects samples from a source domain that are very similar and clustered together.

performance in terms of accuracy (in Fig. 9), it does not start with lowest validation loss, and multiple training epochs are necessary before it converges. The plots show that the training time required by the ORIENT methods to converge is consistently lower than the training time required by Full to converge.

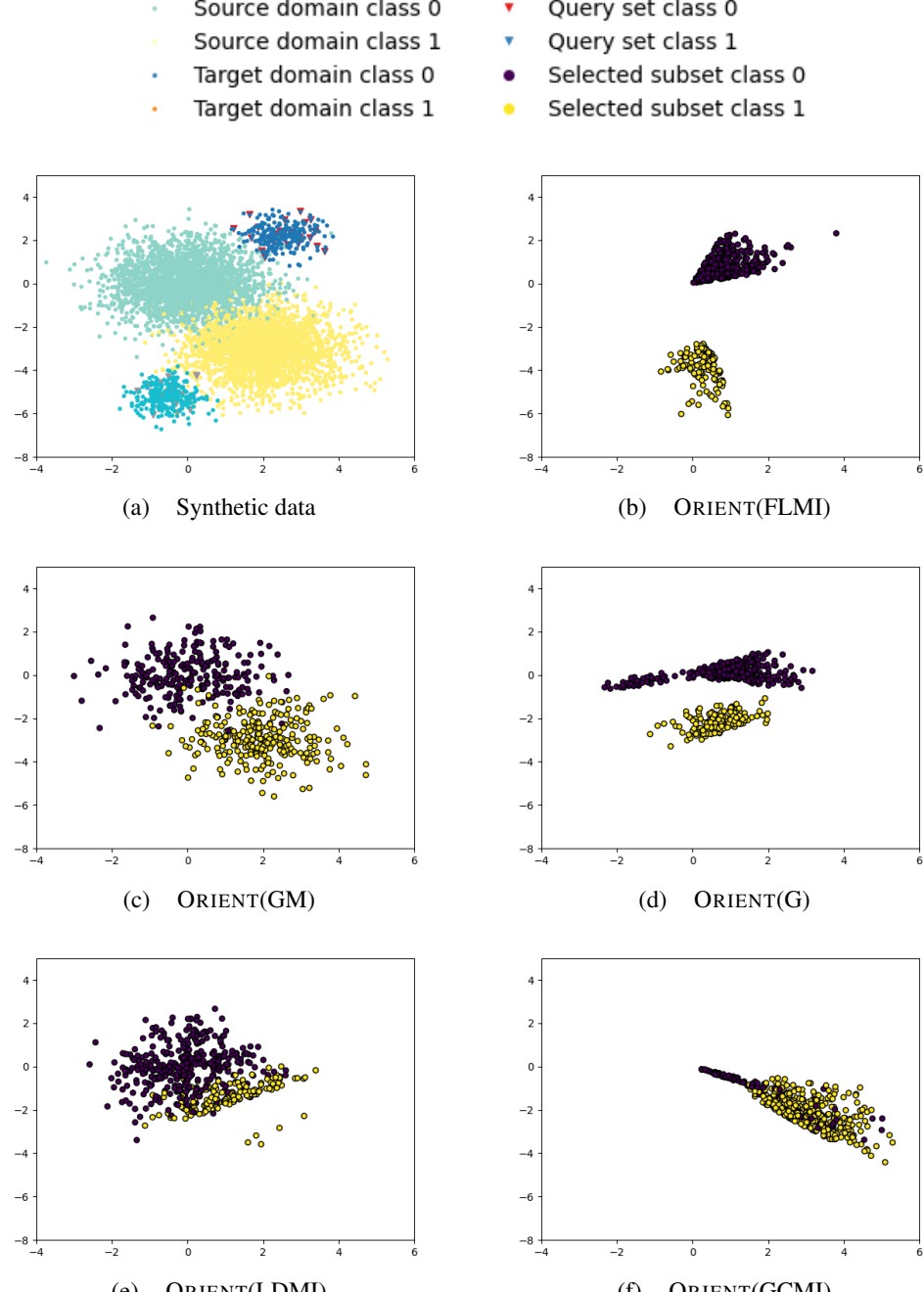

Figure 8: Subsets selected by different instantiations of ORIENT on a synthetic dataset. (a) Synthetic data - We sample 50 examples from the target distributions for the query set. Target domain is skewed to the right for class 0 and left for class 1 as compared to the source domain. (b) ORIENT (FLMI) selects samples close to the target distribution. (c) ORIENT (GM) selects representative samples from the source domain. (d) ORIENT (G) selects samples near the decision boundaries of the source domains. (e) ORIENT(LDMI) prioritizes selection of data samples from certain classes over others (f) ORIENT(GCMI) selects samples from a source domain that are very similar and clustered together.

Tables 16, 17, 18, 19, 20, 21, 22, 23, 24, 25, 26, and 27 present ratio of time taken by subset selection methods with respect to Full training to achieve test accuracy in the range of $(0.3, 0.8)$ with increments of $0.05$. We could see that ORIENT(FLMI) achieves a performance threshold faster

| Method | Epoch Interval (L) | Target domain accuracy | Time Taken(in hrs) | |
|---|---|---|---|---|
| ORIENT(FLMI) | 5 | 0.78 | 3.75 | |
| ORIENT(FLMI) | 10 | 0.78 | 3.61 | |
| ORIENT(FLMI) | 20 | 0.78 | 3.55 | |
| ORIENT(FLMI) | 40 | 0.77 | 3.51 | |

Table 15: Table showing target domain accuracy and training time taken(in hrs) achieved on office-31 (A -> D) using d-SNE loss function and 30% subset.

Table 16: Setting: R → P

| Accuracy threshold | 0.3 | 0.35 | 0.4 | 0.45 | 0.5 | 0.55 | 0.6 | 0.65 | 0.7 | 0.75 | 0.8 |
|---|---|---|---|---|---|---|---|---|---|---|---|
| Random | 6.31 | 6.31 | 6.31 | 6.31 | 6.31 | 6.31 | 6.31 | 3.13 | 0.51 | 0.0 | 0.0 |
| CRAIG | 4.78 | 4.78 | 4.78 | 4.78 | 4.78 | 4.78 | 0.27 | 0.0 | 0.0 | 0.0 | 0.0 |
| ORIENT (FLMI) | 6.17 | 6.17 | 6.17 | 6.17 | 6.17 | 6.17 | 6.17 | 6.17 | 1.92 | 6.21 | 3.21 |
| ORIENT (G) | 2.49 | 2.49 | 2.49 | 2.49 | 2.49 | 2.49 | 2.49 | 2.49 | 0.22 | 2.01 | 0.0 |
| ORIENT (GM) | 3.93 | 3.93 | 3.93 | 3.93 | 3.93 | 3.93 | 3.93 | 1.29 | 0.43 | 3.53 | 0.0 |

Table 17: Setting: R → C

| Accuracy threshold | 0.3 | 0.35 | 0.4 | 0.45 | 0.5 |
|---|---|---|---|---|---|
| Random | 3.22 | 3.22 | 1.62 | 0.0 | 0.0 |
| CRAIG | 0.58 | 0.0 | 0.0 | 0.0 | 0.0 |
| ORIENT (FLMI) | 3.18 | 3.18 | 1.48 | 1.48 | 0.35 |
| ORIENT (G) | 7.58 | 7.58 | 7.58 | 0.72 | 0.45 |
| ORIENT (GM) | 4.83 | 4.83 | 2.04 | 0.97 | 0.3 |

Table 18: Setting: R → A

| Accuracy threshold | 0.3 | 0.35 | 0.4 | 0.45 | 0.5 | 0.55 |
|---|---|---|---|---|---|---|
| Random | 3.38 | 3.38 | 3.38 | 3.38 | 3.38 | 0.0 |
| CRAIG | 4.7 | 4.7 | 4.7 | 4.7 | 0.0 | 0.0 |
| ORIENT (FLMI) | 3.42 | 3.42 | 3.42 | 3.42 | 0.75 | 0.42 |
| ORIENT (G) | 3.4 | 3.4 | 3.4 | 3.4 | 0.31 | 0.26 |
| ORIENT (GM) | 3.35 | 3.35 | 3.35 | 3.35 | 3.35 | 0.9 |

Speedups achieved by different subset selection strategies w.r.t Full training to reach different accuracy thresholds for different combinations using $R$ as source domain on the Officehome dataset in the augmented setting.

than Full in 70 out of 82 cases, ORIENT(GM) achieves a performance threshold faster than Full in 65 out of 82 cases, ORIENT(G) achieves a performance threshold faster than Full in 60 out of 82 cases, whereas CRAIG achieves a performance threshold faster than Full in 27 out of 82 cases and Random achieves a performance threshold faster than Full in 56 out of 82 cases consisting only of lower accuracy thresholds. Furthermore, Random always fails to achieve similar accuracy to Full. From this, it is evident that ORIENTachieves faster performance thresholds than FULL in most cases.

Table 28 presents the ratio of time taken by subset selection methods with respect to Full training to achieve validation loss withing $105\%$ of the minimum validation loss. Even with this impractical stopping criterion we see average speedups of 2.26, 2.37 and 1.85 for ORIENT(FLMI), ORIENT(G) and ORIENT(GM), respectively. Similarly, speedups achieved in reaching $1.1\times$ minimum validation loss for ORIENT(FLMI), ORIENT(G) and ORIENT(GM) are 2.28, 2.38 and 1.85, respectively.

## B  Experiment Details

We use a common training process and hyperparameters for all our experiments. We use the following hyperparameters:

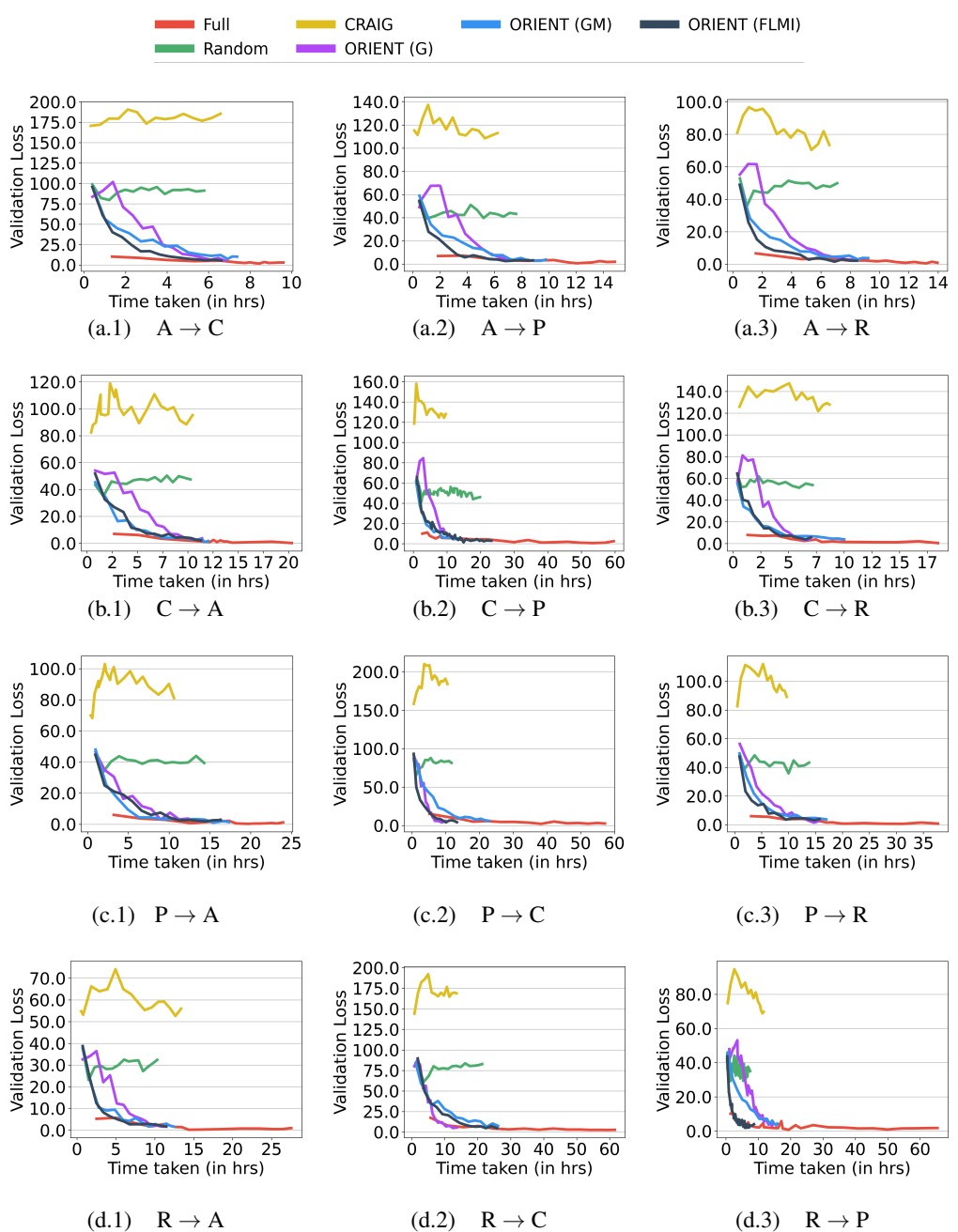

Figure 9: Convergence curves on four domains of Office-Home dataset: Art (A), Clipart (C), Product (P), and Real World (R). $X$-axis presents the training time in hours and $Y$-axis presents the Validation loss on the target domain.

- **Optimization Algorithm:** SGD with Momentum
- **Learning Rate:** 0.001 with Cosine Annealing
- **Momentum:** 0.9
- **Weight Decay:** 5e-4
- **SMI query diversity hyperparameter($\eta$):** 1
- **Number of epochs:** 300

Table 19: Setting: P → R

| Accuracy threshold | 0.3 | 0.35 | 0.4 | 0.45 | 0.5 | 0.55 | 0.6 | 0.65 |
|---|---|---|---|---|---|---|---|---|
| Random | 3.31 | 3.31 | 3.31 | 3.31 | 3.31 | 3.31 | 0.37 | 0.0 |
| CRAIG | 6.24 | 6.24 | 6.24 | 6.24 | 0.39 | 0.0 | 0.0 | 0.0 |
| ORIENT (FLMI) | 3.38 | 3.38 | 3.38 | 3.38 | 3.38 | 3.38 | 0.48 | 1.62 |
| ORIENT (G) | 3.25 | 3.25 | 3.25 | 3.25 | 3.25 | 3.25 | 0.32 | 1.54 |
| ORIENT (GM) | 3.39 | 3.39 | 3.39 | 3.39 | 3.39 | 3.39 | 1.38 | 1.72 |

Table 20: Setting: P → C

| Accuracy threshold | 0.3 | 0.35 | 0.4 | 0.45 | 0.5 |
|---|---|---|---|---|---|
| Random | 9.46 | 9.46 | 4.47 | 0.0 | 0.0 |
| CRAIG | 0.0 | 0.0 | 0.0 | 0.0 | 0.0 |
| ORIENT (FLMI) | 9.72 | 4.02 | 4.02 | 1.48 | Improved |
| ORIENT (G) | 8.88 | 8.88 | 1.27 | 0.91 | 0.0 |
| ORIENT (GM) | 4.02 | 4.02 | 4.02 | 0.47 | 0.0 |

Table 21: Setting: P → A

| Accuracy threshold | 0.3 | 0.35 | 0.4 | 0.45 | 0.5 |
|---|---|---|---|---|---|
| Random | 3.31 | 3.31 | 3.31 | 3.31 | 0.0 |
| CRAIG | 8.32 | 8.32 | 0.0 | 0.0 | 0.0 |
| ORIENT (FLMI) | 3.34 | 3.34 | 1.54 | 0.74 | Improved |
| ORIENT (G) | 3.3 | 3.3 | 3.3 | 0.36 | Improved |
| ORIENT (GM) | 3.28 | 3.28 | 3.28 | 1.39 | Improved |

Speedups achieved by different subset selection strategies w.r.t Full training to reach different accuracy thresholds for different combinations using $P$ as source domain on the Officehome dataset in the augmented setting.

We use a single `GTX 1080 Ti` GPU for experiments.

# C   Code

The code of ORIENT is available at the following link: `https://github.com/athresh/orient`

# D   Licenses

We release the code repository of ORIENT with MIT license, and it is available for everybody to use freely. We use the CORDS—COResets and Data Subset selection—library [23] for the implementation of ORIENT framework and SubModLib—Submodular optimization library—for the SMI optimization. The datasets, Office-31 and Office-Home, are made publicly available by the original authors. Office-Home dataset is made available under a custom license for non-commercial research and educational purposes. Office-Home dataset contains copyrighted material and is made available in accordance with Title 17 U.S.C. Section 107 of the US Copyright Law. In addition, the datasets used do not contain any personally identifiable information.

Table 22: Setting: C → R

| Accuracy threshold | 0.3 | 0.35 | 0.4 | 0.45 | 0.5 | 0.55 | 0.6 |
|---|---|---|---|---|---|---|---|
| Random | 3.38 | 3.38 | 3.38 | 3.38 | 1.69 | 0.0 | 0.0 |
| CRAIG | 2.26 | 2.26 | 0.17 | 0.0 | 0.0 | 0.0 | 0.0 |
| ORIENT (FLMI) | 3.33 | 3.33 | 3.33 | 3.33 | 1.49 | 1.49 | 2.99 |
| ORIENT (G) | 3.29 | 3.29 | 3.29 | 3.29 | 3.29 | 0.24 | 1.81 |
| ORIENT (GM) | 3.29 | 3.29 | 3.29 | 3.29 | 3.29 | 0.5 | 1.54 |

Table 23: Setting: C → P

| Accuracy threshold | 0.3 | 0.35 | 0.4 | 0.45 | 0.5 | 0.55 | 0.6 | 0.65 | 0.7 |
|---|---|---|---|---|---|---|---|---|---|
| Random | 2.22 | 2.22 | 2.22 | 2.22 | 2.22 | 0.44 | 0.0 | 0.0 | 0.0 |
| CRAIG | 9.98 | 9.98 | 0.0 | 0.0 | 0.0 | 0.0 | 0.0 | 0.0 | 0.0 |
| ORIENT (FLMI) | 2.46 | 2.46 | 2.46 | 2.46 | 2.46 | 1.1 | 0.7 | 0.29 | Improved |
| ORIENT (G) | 3.09 | 3.09 | 3.09 | 3.09 | 3.09 | 0.37 | 0.3 | 0.24 | 0.0 |
| ORIENT (GM) | 3.2 | 3.2 | 3.2 | 3.2 | 3.2 | 1.34 | 0.69 | 0.69 | Improved |

Table 24: Setting: C → A

| Accuracy threshold | 0.3 | 0.35 | 0.4 | 0.45 |
|---|---|---|---|---|
| Random | 3.34 | 1.66 | 0.0 | 0.0 |
| CRAIG | 0.0 | 0.0 | 0.0 | 0.0 |
| ORIENT (FLMI) | 3.38 | 3.38 | 0.75 | Improved |
| ORIENT (G) | 3.33 | 1.52 | 0.28 | Improved |
| ORIENT (GM) | 3.34 | 3.34 | 0.89 | Improved |

Speedups achieved by different subset selection strategies w.r.t Full training to reach different accuracy thresholds for different combinations using $C$ as source domain on the Officehome dataset in the augmented setting.

Table 25: Setting: A → R

| Accuracy threshold | 0.3 | 0.35 | 0.4 | 0.45 | 0.5 | 0.55 | 0.6 | 0.65 |
|---|---|---|---|---|---|---|---|---|
| Random | 3.28 | 3.28 | 3.28 | 3.28 | 3.28 | 1.64 | 0.0 | 0.0 |
| CRAIG | 5.26 | 5.26 | 5.26 | 5.26 | 0.45 | 0.29 | 0.0 | 0.0 |
| ORIENT (FLMI) | 3.34 | 3.34 | 3.34 | 3.34 | 3.34 | 3.34 | 1.48 | 0.0 |
| ORIENT (G) | 3.34 | 3.34 | 3.34 | 3.34 | 3.34 | 3.34 | 0.28 | Improved |
| ORIENT (GM) | 3.29 | 3.29 | 3.29 | 3.29 | 3.29 | 3.29 | 1.34 | Improved |

Table 26: Setting: A → P

| Accuracy threshold | 0.3 | 0.35 | 0.4 | 0.45 | 0.5 | 0.55 | 0.6 | 0.65 | 0.7 |
|---|---|---|---|---|---|---|---|---|---|
| Random | 3.77 | 3.77 | 3.77 | 3.77 | 3.77 | 3.77 | 1.8 | 0.0 | 0.0 |
| CRAIG | 15.32 | 15.32 | 2.53 | 0.32 | 0.0 | 0.0 | 0.0 | 0.0 | 0.0 |
| ORIENT (FLMI) | 3.77 | 3.77 | 3.77 | 3.77 | 3.77 | 3.77 | 1.6 | 0.98 | 0.33 |
| ORIENT (G) | 3.76 | 3.76 | 3.76 | 3.76 | 3.76 | 3.76 | 0.37 | 0.37 | 0.24 |
| ORIENT (GM) | 3.74 | 3.74 | 3.74 | 3.74 | 3.74 | 3.74 | 1.41 | 0.5 | 0.36 |

Table 27: Setting: A → C

| Accuracy threshold | 0.3 | 0.35 | 0.4 | 0.45 | 0.5 |
|---|---|---|---|---|---|
| Random | 3.4 | 1.67 | 0.0 | 0.0 | 0.0 |
| CRAIG | 0.0 | 0.0 | 0.0 | 0.0 | 0.0 |
| ORIENT (FLMI) | 3.39 | 1.52 | 1.52 | 0.5 | 0.22 |
| ORIENT (G) | 3.36 | 3.36 | 0.36 | 0.36 | 0.0 |
| ORIENT (GM) | 3.32 | 3.32 | 1.35 | 0.41 | 0.19 |

Speedups achieved by different subset selection strategies w.r.t Full training to reach different accuracy thresholds for different combinations using $A$ as source domain on the Officehome dataset in the augmented setting.

| | C → P | C → R | R → A | P → R | R → P | A → C | C → A | R → C | P → A | A → P | A → R | P → C | Average speedup |
|---|---|---|---|---|---|---|---|---|---|---|---|---|---|
| ORIENT (FLMI) | 3.58 | 2.81 | 1.25 | 2.1 | 3.1 | 1.28 | 1.8 | 2.41 | 1.36 | 1.77 | 1.9 | 3.8 | 2.26 |
| ORIENT (G) | 4.46 | 2.86 | 1.32 | 2.12 | 1.36 | 1.24 | 1.86 | 4.33 | 1.28 | 1.56 | 1.69 | 4.3 | 2.37 |
| ORIENT (GM) | 4.49 | 1.84 | 1.56 | 1.83 | 1.25 | 1.22 | 1.74 | 2.39 | 1.27 | 1.34 | 1.55 | 1.68 | 1.85 |

Table 28: Speed ups achieved by different variants of ORIENT when using $1.05\times$ the minimum validation loss as a stopping criterion for training.