# OpenReview forum: "ORIENT: Submodular Mutual Information Measures for Data Subset Selection under Distribution Shift"
_NeurIPS.cc/2022/Conference — NeurIPS 2022 Accept_

### Official Review · Reviewer_dAxe · 2022-07-08

**Rating:** 7
**Confidence:** 3
**Soundness:** 3 good
**Presentation:** 3 good
**Contribution:** 4 excellent

**Summary:**

This paper introduces ORIENT, a novel subset selection framework to speed up learning in supervised domain adaptation (SDA). Based on submodular mutual information (SMI) functions, ORIENT selects a small diverse subset of training data in the source domain that is similar to data in the target domain. The authors show that ORIENT generalizes the known SDA methods such as GLISTER, GRADMATCH, and a variant of CRAIG. Numerical experiments demonstrated that ORIENT successfully reduces the learning time while achieving the accuracy same as baseline methods.

**Questions:**

- In Algorithm 1, line 4, $\theta$ is updated as $\theta \gets \arg\min \mathcal{L}(A)$. Isn't it a single or a few (stochastic) gradient steps? Otherwise, you get the same $\theta$ for $L$ times until $A$ is changed.

**Strengths And Weaknesses:**

- *Originality*: Novel. This paper unifies the several known methods with SMI functions. To the best of my knowledge, this seems to be the first study in SDA from the viewpoint of SMI functions.
- *Quality*: Good. The presented results seem sound.
- *Clarity*: While I spotted a few typos in the manuscript, the writing is mostly clear.
- *Significance*: This paper has several interesting contributions. I particularly like that the several known methods in SDA are special cases of ORIENT with different submodular functions. The proposed framework can be applied to a wide variety of machine learning models including neural nets, as long as the (stochastic) gradient descent is possible. So adapting ORIENT in the existing ML framework (like PyTorch or TensorFlow) seems quite easy. Furthermore, numerical experiments demonstrate that the proposed method indeed reduces the learning time: sometimes 2-3x speed-ups. I believe that ORIENT is useful and valuable for practitioners.

---

> ### Author Response · Authors · 2022-08-02
> **Response to Reviewer dAxe**
>
> **Q1. In Algorithm 1, line 4,  $\theta$ is updated as $\theta \leftarrow \underset{\theta}{\operatorname{argmin\hspace{0.7mm}}} \mathcal{L}(A)$. Isn't it a single or a few (stochastic) gradient steps? Otherwise, you get the same $\theta$ for $L$ times until $A$ is changed.?**
>
>  We apologize for the confusion. We run mini-batch gradient descent on selected subset/Full data. Hence, the model parameters are updated every epoch. We have updated the algorithm in the rebuttal version of the paper accordingly.

---

### Official Review · Reviewer_mSi3 · 2022-07-11

**Rating:** 6
**Confidence:** 4
**Soundness:** 2 fair
**Presentation:** 3 good
**Contribution:** 3 good

**Summary:**

Authors propose to employ submodular mutual information measures for dataset selection in domain adaptation. Starting with a random subset of data, the proposed algorithm iterates between training on the currently selected dataset and selecting a new dataset based on submodular measure and gradient similarity. Authors claim that the proposed algorithm is much more efficient than full-data training. When combined with d-SNE, ORIENT outperforms full-data training baseline on three adaptation directions and underperforms on three directions, out of 12 directions.

**Questions:**

How do authors compute speedup in Figure 4? I am suspecting authors are fixing the number of passes on the dataset, because speed-up numbers in Figure 4 are very uniform. If so, then I am not sure it is fair to say ORIENT achieves 3x speed-up. FULL could've achieved performance similar with ORIENT when the number of gradient descent steps are the same. Therefore, speed-up should be computed based on how much time it takes to achieve a target performance level.

Can authors provide some discussion on what are main ideas behind proposed SMIs, why they were chosen, what are their conceptual pros & cons, and how do experimental results validate/invalidate these intuitions & assumptions?

How should the size of selected dataset chosen? Can we benefit from more or less frequent dataset selection than authors currently do?

Shouldn't authors compare against dataset selection baselines? Authors' main claim is that their proposed SDA algorithms provide 3x speedup, but maybe we can achieve 3x speedup alternatively by just employing dataset selection on the full data without considering alignment with target domain data.

**Limitations:**

Domain Adaptation via Pseudo In-Domain Data Selection (Axelrod et al, https://aclanthology.org/D11-1033.pdf ) should've been cited, as this paper proposes data selection for domain adaptation. They don't use submodular functions for selection, and they rely on unsupervised model (language model) for selection, so it'd be difficult for authors to compare against Axelrod et al. However, such an experiment would certainly make the paper stronger, by including additional task (machine translation) and comparing against non-submodular baseline (authors don't compare against any non-submodular baseline).

In line 118, authors do not assume monotonicity of submodular function when discussing (1-1/e) approximation guarantee of simple greedy algorithm.

**Strengths And Weaknesses:**

Quality, Significance: The paper builds upon well-established techniques such as submodular mutual information and submodular optimization, which makes the proposed method principled. For example, this allows the selection part of the algorithm to borrow well-known algorithms in submodular optimization literature, and provide approximation guarantee. The proposed framework is also quite general, which will allow future researchers to implement novel algorithms based on it, for example by proposing new similarity measures or mutual information metrics.

Clarity: Authors employ the abstract framework of submodular mutual information, which allows authors to describe their proposed method in high-level. This makes the narrative of the paper quite clear, but this also makes the paper a bit superficial in some of the technical details. For example, it is unclear how four SMIs proposed in this paper have advantages and disadvantages against others. Even intuitive ideas behind these SMIs are not discussed in the main body of the paper. Neither does experiment section discuss how SMIs compare against each other.

Quality: The main contribution of this paper is mostly empirical, as SMIs have been established in prior papers. However, I am not sure  empirical results are very convincing. In Figure 3, Full data selection seems to be quite competitive in most of the adaptation directions even from the first evaluation. Also, there seems to be quite a bit of random errors in this plot, which makes me not sure we can conclude anything other than ORIENT (G), Random, and CRAIG being much worse than others. Results in Table 2 & 3 are mixed. In Table 2, Full outperforms ORIENT 3/6 directions, and ties with ORIENT on 2/6. In table 3, FULL outperforms on 3/12, and ORIENT outperforms on 3/12. Hence we can't really say ORIENT performs better. The only benefit is the reduction of training time, but it is unclear how authors computed the speedup. Therefore, I am not sure ORIENT would be actually computationally more efficient than FULL.

---

> ### Author Response · Authors · 2022-08-02
> **Response to Reviewer mSi3**
>
>
> **Q1. Unclear how four SMIs proposed in this paper have advantages and disadvantages against others. SMI intutions, advantages/disadvantages. How SMIs compare against each other?**
>
>  In Figure 2,3 of the main paper, we demonstrate the performance of ORIENT using different SMI functions on the Office31 and OfficeHome datasets. The results indicate that the ORIENT (GCMI ) performs suboptimally compared to ORIENT using other SMI functions in terms of target domain accuracy. Although ORIENT (LDMI) achieves reasonable target domain accuracy, it is computationally expensive and does not achieve the best performance-speedup trade-off. In comparison to the ORIENT using remaining SMI functions, i.e., ORIENT (GM), ORIENT (G), and ORIENT (FLMI), ORIENT (FLMI) consistently achieves the best performance versus speed-up trade-off.
>
>  Additionally, to provide better intuition into how different SMI functions select data subsets, we present several synthetic experiments in Section A.6 of the updated Appendix. We visualize the subset selected by ORIENT with different SMIs for two toy datasets with $5000$ samples in the source domain and $500$ samples in the target domain using a query set of size $50$. From the results, it is evident that the FLMI function selects sample sources closer to the target domain than other SMI functions. GradMatch function selects representative samples from the source domain. GLISTER selects samples near the decision boundaries of the source domain data. GCMI function selects samples from a source domain that are very similar and clustered together. In synthetic results, we found that the LDMI function tends to prioritize the selection of data samples from certain classes compared to others. Finally, we have also added a section on comparison of SMI functions to the rebuttal version of the paper.
>
> **Q2. ORIENT not performing better than Full?**
>
> Our aim is to reduce training time. With ORIENT we do not aim to out-perform Full in terms of accuracy. Our empirical results show ORIENT provides competitive accuracy while providing significant benefit in training time.
>
> **Q3. How is training time computed?**
>
> We compute the training time taken for a fixed number of epochs (300) and present that in Figure 4. While we can potentially compute the training time required to achieve a particular target performance, in most real-world scenarios, target performance is not known in advance. We also present the learning curves in Figure 3 of the main paper, demonstrating that ORIENT achieves faster convergence than Full. Learning curves also shows that convergence of Full is noisy and thereby resorting to early stopping can lead to sub-optimal performances in some cases (for example Figure 3 (d.1), 3 (d.3), 3(c.3)).
>
> **Q4. Shouldn't authors compare against dataset selection baselines? Authors' main claim is that their proposed SDA algorithms provide 3x speedup, but maybe we can achieve 3x speedup alternatively by just employing dataset selection on the full data without considering alignment with target domain data.**
>
> In our experiments, we compare ORIENT against CRAIG---a coreset selection baseline. CRAIG selects a subset of the source dataset without considering any alignment to the target domain. We have made this explicit in the rebuttal version of the paper. Our results in Figure 3 show that CRAIG indeed achieves speed-up comparable to ORIENT but at the cost of accuracy. ORIENT consistently out-performs CRAIG in terms of accuracy, shown in Figure 2, 3.

---

> > ### Comment · Reviewer_mSi3 · 2022-08-08
> > **Interpretation of Figure 3?**
> >
> > I am not sure how Figure 3 is interpreted as FULL slower than ORIENT. Maybe I am not understanding it well. For example, in Figure 3 (a), Full starts with the highest accuracy even from the first ever evaluation at approximately 1.6 hours. If the validation was run within an epoch (which is not uncommon), I suspect FULL would've achieved even faster convergence than other baselines. An example like (d.3) is maybe a clearer case which ORIENT converges faster than FULL, but I am not sure this is a consistent observation across plots.
> >
> > I understand it's difficult to use a consistent stopping criteria, but using a fixed number of epochs doesn't seem like the right way of demonstrating computational efficiency. For sure, if you use less data, then almost by definition it will take less time to process it 300 times, whereas it is unclear FULL will require the same number of epochs as proposed methods. I really think authors need to come up with a fair stopping criteria to be used across methods. Is validation loss more stable than validation accuracy to be used as a more stable stopping criterion? Or, can you tune patience epochs? Alternatively, you could run the training into convergence, and then regard the time it took to X% of minimal training loss as the training time. I understand that this is not realistic, but it might be OK for the purpose of demonstrating.

---

> > > ### Author Response · Authors · 2022-08-09
> > > **Authors' response**
> > >
> > > We thank the reviewer for engaging in the discussion and highlighting some salient points. We agree that the paper warrants a more thorough discussion of the stopping criterion being used, and we will add more discussion about early stopping in the revised version of the paper.
> > >
> > > Tables 16-27 in the appendix present the speed-ups obtained by various methods with respect to Full to reach different thresholds of test accuracy. We marked the accuracy thresholds that were achieved by subset selection methods and not by Full training as "Improved." Additionally, $0.0$ indicates that the method did not achieve accuracy past that threshold.
> > >
> > > 1) From the speedup results presented in Tables 16-27 of the Appendix, we could see that ORIENT (FLMI) achieves a performance threshold faster than Full in 70 out of 82 cases, ORIENT (GM) achieves a performance threshold faster than Full in 65 out of 82 cases, ORIENT (G) achieves a performance threshold faster than Full in 60 out of 82 cases, whereas CRAIG achieves a performance threshold faster than Full in  27 out of 82 cases and Random achieves a performance threshold faster than Full in 56 out of 82 cases consisting only of lower accuracy thresholds. Furthermore, Random always fails to achieve similar accuracy to Full. From this, it is evident that ORIENT achieves faster performance thresholds than FULL in most cases.
> > >
> > > 2) As the reviewer pointed out, there are indeed some cases where Full achieves high classification accuracy very quickly. However, this was only in terms of test accuracy, and the loss curves shown in Figure 9 of the Appendix indicate that the model has not yet converged. In preliminaries, we mentioned that domain adaptation loss is comprised of both cross-entropy loss and feature alignment loss. Due to feature alignment loss, although the model achieved high accuracy very quickly, it does not necessarily mean convergence in terms of training loss.
> > >
> > > 3) Additionally, despite the fact that the accuracy curves for Full training are not monotonic, the loss curves for Full training (Figure 9 in the Appendix) are monotonic. An interesting observation is that with ORIENT, the accuracy and loss curves are both monotonous. We speculate that the reason is that ORIENT selects source domain subsets that are similar to the target domain for training and improves feature alignment and class discrimination simultaneously, whereas full training does not.
> > >
> > > **Q.  I really think authors need to come up with a fair stopping criteria to be used across methods. Is validation loss more stable than validation accuracy to be used as a more stable stopping criterion. Alternatively, you could run the training into convergence, and then regard the time it took to X% of minimal training loss as the training time. I understand that this is not realistic, but it might be OK for the purpose of demonstrating.**
> > >
> > > We present the speedups achieved in reaching $1.05 \times$ minimum validation loss for each of the methods across all the source-target combinations on the Office-Home dataset in Table 28 in the Appendix.
> > >
> > > 1) From the results, We see average speedups of 2.26, 2.37 and 1.85 for ORIENT (FLMI), ORIENT (G) and ORIENT (GM), respectively. Similarly, speedups achieved in reaching $1.1 \times$ minimum validation loss for ORIENT (FLMI), ORIENT (G) and ORIENT (GM) are $2.28$, $2.38$ and $1.85$, respectively.
> > >
> > > 2) We see that ORIENT (FLMI) achieves speedups of around $2.26$ times compared to Full even while using $x%$ of minimum validation loss (we consider 105\% and 110\%) - an impractical stopping criterion, as acknowledged by the reviewer.
> > >
> > > 3) We also note that tuning patience epochs for Full is not straight-forward in our experiments. Notice that accuracy of Full, in plots (a.3), (b.1), (c.2), (d.1) and (c.3) in Figure 3, for example, goes down for several epochs before achieving maximum accuracy. In plot (c.3), Full achieves high accuracy after 5 hours of training but requires another 20 hours of training to match that accuracy before surpassing it. This renders setting patience epochs across source-target combinations infeasible. Additionally, we observe that for Full training validation loss continues to decrease steadily even as test accuracy decreases for several epochs.

---

> ### Author Response · Authors · 2022-08-02
> **Response to Reviewer mSi3, continued**
>
>
>  **Q5. How is the size of selected dataset chosen?**
>
> We present comparison of target domain accuracy achieved by ORIENT for two data subset sizes of 10% and 30% on Office31 dataset in the table below. Results demonstrate that the data subset size triggers a trade-off between the target domain accuracy and speed up. Larger data subset sizes improves the model performance while compromising on efficiency (i.e., increased training time), whereas, small data subset sizes improves the efficiency while compromising on performance. In our experiments, we considered 30% data subset as ORIENT achieves similar performance to FULL using 30% subset while being 3x faster. We have also added the detailed analysis with subset size to Section A.7 of the Appendix.
>
>
> |                | Fraction | A $\rightarrow$ D | A $\rightarrow$ W | D $\rightarrow$ A | D $\rightarrow$ W | W $\rightarrow$ A | W $\rightarrow$ D |
> |----------------|----------|-------------------|-------------------|-------------------|-------------------|-------------------|-------------------|
> | Full           | 1.0      | $0.77$            | $0.69$            | $0.53$            | ${0.93}$          | $0.54$            | ${0.98}$          |
> | Random         | 0.1      | $0.65$            | $0.58$            | ${0.43}$          | $0.62$            | $0.48$            | ${0.72}$          |
> | Random         | 0.3      | $0.76$            | $0.68$            | $0.53$            | $0.86$            | $0.53$            | $0.94$            |
> | ORIENT (FLMI) | 0.1      | $0.70$            | $0.67$            | ${0.50}$          | $0.77$            | $0.48$            | ${0.92}$          |
> | ORIENT (FLMI) | 0.3      | $0.78$            | $0.71$            | ${0.55}$          | $0.90$            | $0.56$            | ${0.97}$          |
> | ORIENT (G)    | 0.1      | $0.75$            | $0.60$            | ${0.42}$          | $0.70$            | $0.48$            | ${0.89}$          |
> | ORIENT (G)    | 0.3      | $0.76$            | $0.66$            | ${0.50}$          | $0.87$            | $0.52$            | ${0.96}$          |
>
> Above table shows target domain accuracy achieved on office-31 using d-SNE loss function for different fractions of subset.
>
>
>
> |                 | Fraction | A $\rightarrow$ D | A $\rightarrow$ W | D $\rightarrow$ A | D $\rightarrow$ W | W $\rightarrow$ A | W $\rightarrow$ D |
> |-----------------|----------|-------------------|-------------------|-------------------|-------------------|-------------------|-------------------|
> | Full            | 1.0      | $11.57$           | $8.33$            | $2.01$            | $2.21$            | $2.18$            | $3.42$            |
> | Random          | 0.1      | $1.18$            | $0.83$            | $0.20$            | $0.22$            | $0.23$            | $0.34$            |
> | Random          | 0.3      | $3.49$            | $2.48$            | $0.61$            | $0.65$            | $0.69$            | $1.04$            |
> | ORIENT (FLMI) | $0.1$    | $1.25$            | $0.89$            | $0.25$            | $0.26$            | $0.26$            | $0.38$            |
> | ORIENT (FLMI) | $0.3$    | $3.55$            | $2.53$            | $0.66$            | $0.69$            | $0.72$            | $1.06$            |
> | ORIENT (G)    | $0.1$    | $1.21$            | $0.87$            | $0.25$            | $0.26$            | $0.25$            | $0.37$            |
> | ORIENT (G)    | $0.3$    | $3.53$            | $2.51$            | $0.66$            | $0.69$            | $0.72$            | $1.06$            |
>
> Above table shows training time in hours on Office-31 with d-SNE loss function for different fractions of subset.

---

> ### Author Response · Authors · 2022-08-02
> **Response to Reviewer mSi3, continued**
>
>
> **Q6. Can we benefit from more or less frequent dataset selection than authors currently do?**
>
> In the table below, we present comparison of target domain accuracy achieved by ORIENT (FLMI) for different $L$ values of $5, 10, 20, 40$ on Office 31 (A->D) using d-SNE loss and 30% subset. Note that smaller the value of $L$ is, greater the frequency of subset selection. Results demonstrate that using $L=5, 10$ (i.e., more frequent subset selection) results in higher training time with no improvement in accuracy. Whereas using $L=40$ (i.e., less frequent subset selection) results in lower target domain accuracy with not much significant improvement in training time. Hence, we used $L=20$ in our experiments.  We have also added the detailed analysis with $L$ to Section A.8 of the Appendix.
>
> | Method         | Epoch Interval (L)   | Target domain accuracy | Time Taken(in hrs)|
> |----------------|----------------------|------------------------|-------------------|
> | ORIENT (FLMI) |             5        |      $0.78$            | $3.75$            |
> | ORIENT (FLMI) |             10       |      $0.78$            | $3.61$            |
> | ORIENT (FLMI) |             20       |      $0.78$            | $3.55$            |
> | ORIENT (FLMI) |             40       |      $0.77$            | $3.51$            |
>
> Above table showing target domain accuracy and training time taken(in hrs) achieved on office-31 (A -> D) using d-SNE loss function and 30% subset.
>
> **Q7. Related work: Axelrod et al, https://aclanthology.org/D11-1033.pdf**
>
>   Thanks for your suggestion. We have cited the paper in the related work section of the revised paper.
>
> **Q8. In line 118, authors do not assume monotonicity of submodular function when discussing (1-1/e) approximation guarantee of simple greedy algorithm.**
>
> We apologize for the typo in the paper. All Submodular mutual information measures except for LogDetMI are monotone submodular functions and hence the greedy approximation guarantees of 1-1/e holds true. Even though LogDetMI is not submodular, previous works [1, 2] have reported good empirical performance using the lazy greedy algorithm for maximization of the LogDetMI function. Following the previous works [1, 2], we also use the lazy greedy algorithm to maximize the LogDetMI function. We modified the typo and expanded upon this in the rebuttal revision of the paper in lines 162-167.
>
> 1. Suraj Kothawade, Nathan Beck, KrishnaTeja Killamsetty, Rishabh K. Iyer: SIMILAR: Submodular Information Measures Based Active Learning In Realistic Scenarios. NeurIPS 2021: 18685-18697
> 2. Suraj Kothawade, Vishal Kaushal, Ganesh Ramakrishnan, Jeff A. Bilmes, Rishabh K. Iyer: PRISM: A Rich Class of Parameterized Submodular Information Measures for Guided Data Subset Selection. AAAI 2022: 10238-10246

---

### Official Review · Reviewer_HJ5G · 2022-07-11

**Rating:** 7
**Confidence:** 3
**Soundness:** 4 excellent
**Presentation:** 4 excellent
**Contribution:** 4 excellent

**Summary:**

For SDA (Supervised Domain Adaptation), authors propose a framework to select a subset from source domain data so that the training process can be speeded up and better performance can be achieved.

**Questions:**

Can this framework be extended to UDA?


**Limitations:**

Stated in weaknesses.


**Strengths And Weaknesses:**

Strengths:

- The problem setting is novel, interesting and realistic. It is great to see ideas focusing on utilizing data instead of building fancy models which provides marginal improvement of performance.

- Comprehensive experiments are conducted to demonstrate the effectiveness of this framework.

- The paper is well-written and presented.

Weaknesses:

- The calculation of the similarity matrix is quite straightforward. Can it be learned or calculated more properly?

---

> ### Author Response · Authors · 2022-08-02
> **Response to Reviewer HJ5G**
>
> **Q1. Calculation of the similarity matrix is quite straightforward. Can it be learned or calculated more properly?**
>
> Our empirical evaluations show that the standard cosine similarity works for image data sets. We will consider more sophisticated similarity measure in our future work.
>
> **Q2. Can this framework be extended to UDA?**
>
> We can extend our approach to UDA by using hypothesized labels for computation of loss and gradients. One other approach is to use feature embeddings, instead of depending on gradients, for computing similarity matrix used by SMI functions.

---

> > ### Comment · Reviewer_HJ5G · 2022-08-09
> > **Thanks for your response.**
> >
> > Thanks for your response. After reading the response and other reviews, I keep my score.

---

### Official Review · Reviewer_SGGh · 2022-07-12

**Rating:** 3
**Confidence:** 4
**Soundness:** 2 fair
**Presentation:** 3 good
**Contribution:** 2 fair

**Summary:**

It is proposed to speed up the existing SDA methods by selecting a subset of the source dataset that is highly "similar" to the target dataset and training only using the source subset. Since model training is performed on smaller set (given budget), the computational effort is lessened. Also, since the subset is more aligned to the target SDA is explicitly performed.

Subset similarity is measured through a SMI, with pairwise data similarity defined as cosine similarity of the gradient of loss at the points wrt. the model parameter.

The overall algorithm alternates between loss optimization (training/learning) and optimal subset selection under SMI.

Using simulations on benchmarks it is argued that computational efficiency is achieved without much compromise on accuracy.

**Questions:**

1. How does the initialized subset effect the final accuracy? Can this be experimentally studied via simulations?

2. Since the initialization of subset is random, reporting mean and std div over multiple initializations seems a better way of presenting the results.


**Limitations:**

Memory limitation is mentioned. However some of the above written limitations can be discussed in more detail.

**Strengths And Weaknesses:**

The basic strategy of restricting training to merely a subset inherently seems to have some limitations:

1. The valuable information in the complement of the selected subset is completely ignored.
2. The extent of complement's usefulness is not only determined by its similarity to the target set. Perhaps, the complement can be used in other subtle ways: for example training domain-invariant features etc.
However, such aspects and corresponding trade-off have not been discussed.

Since the proposed methodology explicitly performs SDA, it seems very critical to empirically compare against few state-of-the-art SDA approaches. Especially, those which are scalable. However, the simulations do not seem to compare even wrt one SDA method. the baselines are simply other alternative subset methods like random/core, and training with full dataset (which is also NOT an SDA method). Without these critical comparisons the practical usefulness of the method cannot be established.

----------
rebuttal resolved some concerns. thanks.

---

> ### Author Response · Authors · 2022-08-02
> **Response to Reviewer SGGh**
>
> **Q1. The basic strategy of restricting training to merely a subset inherently seems to have some limitations**
>
>  Orient is an adaptive subset selection process, so throughout the model training, the subsets keep on changing. Therefore, the samples that are never seen by the training model is usually very small. For instance, in the office home domain, with 30% subset selection, the percentage of samples never selected is between 3%--8% for ORIENT (FLMI). For random, however, the percentage of samples never selected is between 47%--49%. Furthermore, the samples that are never selected during model training are either redundant or significantly different from target distribution.
>
> **Q2. Since the proposed methodology explicitly performs SDA, it seems very critical to empirically compare against few state-of-the-art SDA approaches. Especially, those which are scalable. However, the simulations do not seem to compare even wrt one SDA method. the baselines are simply other alternative subset methods like random/core, and training with full dataset (which is also NOT an SDA method). Without these critical comparisons the practical usefulness of the method cannot be established.**
>
> We present empirical evaluation against two state-of-the-art SDA approaches: d-SNE and CCSA, in Table 2, 3, 7, 8, 11, 12. ORIENT is a subset selection approach which can be used in complement with any existing SDA approaches. Hence, we evaluate ORIENT against Full dataset, random subset and coreset appraoches, with standard as well as state-of-the-art SDA loss functions.
>
> **Q3. How does the initialized subset effect the final accuracy? Can this be experimentally studied via simulations?**
>
> In our experiments, we randomly select the initial subsets. Our results, in Tables 5 and 7, show std dev is less than 0.04 in most of the experiments, so the impact of the initial subset on the final accuracy is minimal.
>
> **Q4. Since the initialization of subset is random, reporting mean and std div over multiple initializations seems a better way of presenting the results.**
>
> We use 5 random initialization for all the results. We present mean and std dev in the Appendix, Tables 5, 6, 7, 8.

---

> > ### Comment · Reviewer_SGGh · 2022-08-08
> > **thanks**
> >
> > My major concerns seem to have been addressed. It seems I misunderstood a few things during initial review. I will increase my score a bit accordingly. However, I feel the contribution and novelty are a limited.

---

> > > ### Author Response · Authors · 2022-08-08
> > > **Response to Reviewer SGGh**
> > >
> > > Thank you for your response. We are glad that your initial major concerns are now addressed. However, your rating suggests that you still find our paper has technical flaws and weak evaluation. If you could elaborate on what are the technical flaws you are concerned with, we would be more than happy to address those.

---

### Meta-Review · Area_Chair_s1fd · 2022-08-26

**Recommendation:** Accept
**Confidence:** Certain

**Metareview:**

The paper contributes to an important research direction: reducing the computing resources needed for training deep learning models while achieving state-of-the-art results. The proposed method is a principled strategy for domain adaptation problems. While based on existing notions, they are used cleverly. The diverse and extensive experiments show that relying on submodular mutual information to select target points is a promising strategy (although reporting the error bars more systematically would be most appreciated). The activation maps are a nice addition to commonly reported accuracy metrics.

For the reasons mentioned above, I recommend accepting the paper.

I strongly encourage the authors to consider the reviewer's discussion to provide an improved version. In addition to reviewers' suggestions for improvements, I would like to see in the revised version:
- Line 118: set operations $A \cup x$ written as $A \cup \lbrace x \rbrace$ ;
- Figures 3 and 4: error bars obtained by repeating the experiments with random data splits ;
- Line 278: the name of the missing author (H. Shimodaira) to reference [1] ;
- Section A.5: a clear explanation of how are computed the standard deviations reported in Tables 5,6,7,8.

**Award:**

No

---

### Decision · Program_Chairs · 2022-09-14

Accept